# A systematic approach to the scale separation problem in the development of multiscale models

**Pinaki Bhattacharya**[1,2]*, **Qiao Li**[1,2], **Damien Lacroix**[1,2], **Visakan Kadirkamanathan**[2,3], **Marco Viceconti**[4,5]

**1** Department of Mechanical Engineering, University of Sheffield, Sheffield, United Kingdom, **2** INSIGNEO Institute for *in silico* Medicine, University of Sheffield, Sheffield, United Kingdom, **3** Department of Automatic Control and Systems Engineering, University of Sheffield, Sheffield, United Kingdom, **4** Dipartimento di Ingegneria Industriale, Alma Mater Studiorum – University of Bologna, Bologna, Italy, **5** Laboratorio di Tecnologia Medica, IRCCS Istituto Ortopedico Rizzoli, Bologna, Italy

* p.bhattacharya@sheffield.ac.uk

**Data Availability Statement:** The original subject-specific data used to illustrate the models was generated as part of the VPHOP EU-FP7 project (grant number 223865). It was derived from

## Abstract

Throughout engineering there are problems where it is required to predict a quantity based on the measurement of another, but where the two quantities possess characteristic variations over vastly different ranges of time and space. Among the many challenges posed by such 'multiscale' problems, that of defining a 'scale' remains poorly addressed. This fundamental problem has led to much confusion in the field of biomedical engineering in particular. The present study proposes a definition of scale based on measurement limitations of existing instruments, available computational power, and on the ranges of time and space over which quantities of interest vary characteristically. The definition is used to construct a multiscale modelling methodology from start to finish, beginning with a description of the system (portion of reality of interest) and ending with an algorithmic orchestration of mathematical models at different scales within the system. The methodology is illustrated for a specific but well-researched problem. The concept of scale and the multiscale modelling approach introduced are shown to be easily adaptable to other closely related problems. Although out of the scope of this paper, we believe that the proposed methodology can be applied widely throughout engineering.

## 1. Introduction

In the decade 2010–2019, Google Scholar indexed 16,600 publications that contained the word "multiscale" in the title, more than twice of those in the previous decade (8,190). This shows how multiscale modelling is becoming increasingly popular. As the interest in this field has increased, so has the sophistication of the methods involved. However, one key aspect has remained scarcely investigated.

In physics, the term "scale" is used to indicate a range of possible length or time values, in close relation with the "order of magnitude" concept. Multiscale modelling becomes necessary

clinical imaging data collected during a previous research project, funded by the Medical Research Council, UK (grant number G0601272), and the National Institute for Health Research (NIHR), UK. The original data are restricted by The University of Sheffield Ethics Committee and available only in the frame of collaboration agreements with other research institutions. Please contact the corresponding author for further information on access policies for the original data. The minimal anonymised processed data needed to reproduce the results reported in this paper are freely downloadable from the following URL: https://doi.org/10.15131/shef.data.12284306.

**Funding:** 1. Grant number: EP/K03877X/1 (Project MultiSim) Funder name: UK Engineering and Physical Sciences Research Council Funder URL: https://epsrc.ukri.org Authors receiving funding: PB, DL, VK, MV. 2. Grant number: EP/S032940/1 (Project MultiSim2) Funder name: UK Engineering and Physical Sciences Research Council Funder URL: https://epsrc.ukri.org Authors receiving funding: PB, VK. 3. Grant number: H2020-EINFRA-2015-1/675451 (CompBioMed Centre of Excellence) Funder name: European Commission H2020 programme Funder URL: https://ec.europa.eu/programmes/horizon2020/en Authors receiving funding: MV. 4. Grant number: H2020-INFRAEDI-2018-1/823712 (CompBioMed Centre of Excellence2) Funder name: European Commission H2020 programme Funder URL: https://ec.europa.eu/programmes/horizon2020/en Authors receiving funding: PB, MV. 5. Grant number: H2020-WIDESPREAD-2018-01/857533 (SANO European Centre for Computational Medicine) Funder name: European Commission H2020 programme Funder URL: https://ec.europa.eu/programmes/horizon2020/en Authors receiving funding: PB, MV. 6. Grant number: IS-BRC-1215-20017 (NIHR Sheffield Biomedical Research Centre - Translational Neuroscience) Funder name: UK National Institute for Health Research Funder URL: https://www.nihr.ac.uk/ Authors receiving funding: MV. 7. Grant number: none Funder name: Shun Hing Education and Charity Fund Funder URL: http://www.shunhinggroup.com Authors receiving funding: QL.

**Competing interests:** The authors have declared that no competing interests exist.

when the phenomenon to be modelled manifests across such a wide space–time range that is impossible to capture it with a single model. In this situation, this range is subdivided into several scales in a process that is hereinafter referred to as "scale separation" [1]. The phenomenon is then modelled at each scale, and these single-scale models are finally linked to form a multiscale model. While there is considerable methodological interest in the methods used to link models across space–time scales, there is little attention to the process through which we decide the appropriate scale separation for a given problem.

In some applications, such as condensed matter mechanics, scale separation is chosen in relation to the range of validity of the idealisations used to build each single scale model. Thus, we model at one scale using quantum mechanics, at another using molecular dynamics, at yet another using dislocation theory (solids) or dissipative particle dynamics (fluids) and last using continuum mechanics [2–4].

In some other applications scale separation is a mathematical construct. This approach is common when one physical idealisation is valid (or assumed to be valid) over the entire space–time range. In such a situation, multiscale modelling can leverage some level of self-similarity, space–time periodicity or macrohomogeneity present in the phenomenon of interest [3,5–8]. Multiscale modelling approaches that employ this notion of scale separation are sometimes termed as applied mathematics and continuum mechanics or continuum micromechanics approaches (henceforth referred to as AM&CM for brevity). The motivation for using a multiscale approach is computational efficiency [9,10]. Self-similar systems are usually characterised by a very large number of scales. Solving the scale invariant equations allows to predict the behaviour of the whole system with significantly less expense [11,12]. In systems with periodicity or macrohomogeneity, the size of the representative volume element (RVE) defines the characteristic finer scale. It should be noted that identifying the RVE size is not always straightforward, especially in the presence of material randomness [10,13]. In practical uses of this approach, the requirement for asymptotic scale separation (e.g. ratio of characteristic 'scales') can in some instances be relaxed to as low as a factor of 3 [14]. Not surprisingly, AM&CM approaches have been useful in providing key insights in areas such as bone mechanics [15,16] where self-similarity, space–time periodicity or macrohomogeneity are perhaps non-trivial assumptions.

In applications dealing with man-made systems—such as in systems engineering [17–19]– scale separation is implicit in the hierarchical organisation of components comprising the large system, such as a big aircraft. Such organisation can relate to arrangement in space–time, or to flow of information between components.

In living organisms, the subdivision in parts is not clear, nor is the hierarchy of organisation well-defined. Yet, in multiscale models of living organisms, the scale separation employed is commonly based on descriptive anatomy, and this choice of basis is often assumed to be a "natural" one. Thus, most multiscale models in biomedicine are separated into scales using anatomical concepts such as "body", "organ", "tissue", "cell", etc. [20–22]. Although such models are sometimes analysed using techniques that are common to AM&CM based modelling approaches, the definition of scale employed here is quite different. Unfortunately, this is probably the least rigorous of all the multiscale approaches described above; as for example considering the organs in the human body, the liver weighs 1.7 kg, the pineal gland 170 g. The range in size is even bigger: the longest bone is the femur (500–600 mm), the shortest is the stapes in the ear (3 mm). So, the concept of "organ" is not well defined dimensionally, and the same holds for any other anatomical concept. Worse, in many studies it is implicitly assumed that the temporal scales are separated consistently with the spatial ones; for example, the life span range is associated to the whole-body scale, while the milliseconds range is associated to the molecular scale. However, there is very little evidence to support such an assumption. For

example, a nerve impulse to an extrafusal muscle fibre can travel half of the body in less than 100 ms and ocular saccades occur in 20–30 ms.

Even without a comprehensive review of any of the above approaches, one can observe that the meaning of the terms 'multiscale modelling', 'scale' and 'scale separation' depend on which approach is used. In some areas of application, there exist problems of a multiscale nature where only one idealisation (e.g. continuum mechanics) sufficiently captures all phenomena of interest. Yet, either the assumption of self-similarity, space–time periodicity or macrohomogeneity is not fit for purpose and/or its nature is challenging to characterise precisely. Biomedicine is a typical example of such application, where either the natural variability does not lend itself to abstractions such as self-similarity, periodicity or macrohomogeneity, or ethical considerations hinder the precise characterisation of the nature of such abstractions, or both. In addition, current methods of defining scale and scale separation based on descriptive anatomy are ambiguous. There might be other application areas, that the authors are unaware of, with similar challenges. In response to this challenge, this study proposes a systematic approach to scale separation based on the concepts of "grain" and "extent". These concepts relate to the space/time resolution of the instrumentation used to observe the phenomenon of interest, and the size/duration of the features of interest in that phenomenon. This approach is illustrated by considering an exemplar problem: forecasting the future mechanical strength of a whole bone due to continuous biochemical processes such as bone remodelling taking place inside it. This phenomenon will be first modelled mathematically at a single scale, assuming infinite resolution (i.e. without any limitation on observable detail). The impossibility of solving such a model will be discussed and the scale separation method proposed above will be attempted to construct a multiscale model to solve the problem. The versatility of the multiscale modelling method will then be explored by varying the set of instrumentations used to inform the model. This versatility will be further investigated by applying the approach to a different multiscale problem.

## 2. Method

### 2.1. Model description: Need for a multiscale approach

The description of a scientific model comprises: the specific question to be answered, the portion of reality to be considered and the details of the abstraction process. These details are given below in somewhat generic terms for an exemplar problem, in order to avoid implying any loss of generality of the multiscale approach that follows the model description.

The specific question to be answered is as follows. Consider a bounded volume that is a solid continuum. Without loss of generality, this volume is identified henceforth with the bone organ. If at time $t = T^*$, a load and several constraints are applied on the boundary of this bone in a specific configuration, what is the minimum magnitude of the load that will lead the bone to mechanical failure?

Only the following portion of reality is considered to be relevant to answer the above question. The bone $B$ occupies a connected volume $\Omega$ in $\mathbf{R}^3$, is bounded by the surface $\Gamma$ and no two points in $\Omega$ are separated by a distance larger than $L^*$. The mechanical response of $B$ at any point of time depends on its instantaneous material composition. The mechanical response of $B$ changes over time due to biochemical processes occurring within it. The bone $B$ exchanges mechanical and biochemical energy with the rest of the universe $B'$. This exchange is essential to sustain the relevant biochemical processes occurring within $B$. Note that, in the duration $0 \leq t < T^*$, the exchange of mechanical and biochemical energy leads to instantaneous loads on its boundary $\Gamma$. It is assumed that these loads do not lead to a mechanical

failure; this assumption is required to ensure that the original question, whether the hypothetical load applied at $t = T^*$ will lead to mechanical failure, remains sensible.

The portion of reality described above can be expressed mathematically. This requires abstracting characteristics of this reality in terms of quantifiable variables. It is assumed that for all quantities relevant to the problem, one can ignore variations separated by distances $l^*$ or smaller and variations separated by durations $t^*$ or shorter. The mathematical expressions are not detailed here, as these will be specific to the application being considered. The consideration of relevance here is the case when either (or both) the maximum and minimum distances and time spans of interest are very disparate, i.e. $l^* \ll L^*$ and $t^* \ll T^*$. In particular, the disparity is considered to be so large that (a) obtaining numerical solutions to the mathematical equations of the model is computationally prohibitive, and/or (b) no single experiment can completely characterise the bone B in terms of the input and output variables of the model over such a large space–time range. The curse of resolution gives rise to such a situation and motivates a multiscale modelling approach [1].

## 2.2. Outline of the multiscale modelling approach

In order to make it possible to analyse the model, its large spatiotemporal domain of interest needs to be decomposed into multiple *scales*. Here, scale is defined in terms of two attributes: *grain* and *extent*, in both space and time. Grain is defined as the larger of the minimum distance (or time span) that can be distinguished by the instrumentation, or as the characteristic distance (or time span) of variation of the smallest (or fastest) feature of interest measured using this instrumentation. Extent is defined as the smaller of the maximum distance (or time span) that can be measured by the same instrumentation, as the characteristic distance (or time span) of variation of the largest (or slowest) feature of interest measured using this instrumentation. In the following, the spatial and temporal grains for a scale S are denoted respectively by $l^*_S$ and $t^*_S$, and the corresponding extents are denoted by $L^*_S$ and $T^*_S$.

The model described in the previous section possesses a *hypothetical* single scale that is defined by the grains $l^*$ and $t^*$ and the extents $L^*$ and $T^*$. This scale is hypothetical because only hypothetical instruments possessing infinite resolution (i.e. unlimited observable detail) can fully characterise the portion of reality situated within this scale. To determine real scales, one needs to find real instrumentations with finite lower and upper limits that overlap with the maximum and minimum distances and time spans of interest of the hypothetical scale. In addition, one requires empirical evidence of smallest, largest, slowest and fastest features of interest that can be measured by these instrumentations. This exercise is carried out to the point that the hypothetical scale of the problem has been sufficiently populated with real scales $S_1, S_2 \ldots S_n$. For each real scale, the mathematical equations describing the original problem are revised, with reinterpretations as necessary for each variable of the original problem. The multiscale model is closed by adding equations to relate variables at different scales. At this point, an algorithmic orchestration to solve the multiscale model can be developed.

## 2.3. An illustration of the multiscale modelling approach

The multiscale modelling approach outlined above is strongly tied to real features and real instrumentations. Hence, the model described earlier in §2.1 in somewhat generic terms, is now specified in more detail by referring to a real problem. First, the specific question to be answered is reformulated: what will be the side-fall strength of a subject's femur after 10 years from now? Assessment of hip fracture risk over a 10-year period is of significant clinical interest [23]. Femur strength is a strong predictor of hip fracture risk [24]. Side-fall strength refers to the particular case where the force is applied on the surface of the femur head and

constraints are applied at the greater trochanter and at the distal end of the femur. In our approach, the strength of the femur is defined as the smallest magnitude of force that needs to be applied to cause it to fail mechanically, as we vary the impact angles.

**2.3.1. Description of the closed system.** In this section the description of the relevant portion of reality is given in more detail. The portion of reality considered here is a much smaller subset of the current body of scientific knowledge, relative to what would be undertaken in a typical multiscale modelling research study. This choice is motivated by the desire to keep the illustration brief. Yet, as the examples in the Supporting Information section show, the multiscale modelling approach detailed in this paper can readily be applied when brevity is not a constraint.

Consider a femur B, occupying a connected volume $\Omega$ in $\mathbf{R}^3$, bounded by the surface $\Gamma$ and possessing a maximum size of 0.44 m [average length of the adult human femur 25, 26]. The femur is contained fully and at all times within a human subject's body over the duration $0 \leq t \leq 10$ years. It is assumed that over this duration, the subject remains alive with sufficient metabolic rate such that mechanical and biochemical energy is exchanged continuously between the femur and the rest of the body; yet, the mechanical energy exchange never exceeds the level that would lead the femur to fracture.

The femur volume is assumed to comprise a single material phase that fails in brittle fashion at small strains when loaded at physiological strain rates. Impact forces applied on the femur surface (such as during a fall to the side) and strains within the femur possess characteristic temporal variations in the range of 1–1000 ms [27,28]. Below the failure strain, the material response depends on the local volumetric bone mineral density (vBMD), which is distributed heterogeneously throughout the femur volume. Local variations in vBMD, as measured using clinical quantitative computed tomography (qCT), can capture underlying features such as thin cortices in the femur neck [of the order of 0.5 mm, see 29–31]. Heterogeneity of vBMD distribution is important for predicting strains in the femur [32]. Due to biochemical activity occurring within the bone, progressive changes occur in femur strength characteristically over durations of 1 year [33]. For simplicity, henceforth, distances and time spans defining the lower and upper bounds of a portion of reality are rounded down and rounded up, respectively, to the nearest power of 10. Thus, the spatiotemporal domain of observation ranges between $l^* = 10^{-4}$ m to $L^* = 10^0$ in length and from $t^* = 10^{-3}$ s to $T^* = 10^9$ s in time.

**2.3.2. Abstraction of the closed system.** The material phase of the femur is assumed to possess a spatially heterogeneous, rate-independent, isotropic linear elastic response with failure at small strains. Spatial heterogeneity is based on the well-known disparity of mechanical response between cancellous and cortical regions of bone [34]. Evidence of strain-rate independence for strain-rates corresponding to fall loading scenarios is well-known from experiments [35–38]. Assumption of isotropic linear elasticity is justified by the accuracy of prediction of bone strength by finite-element models based on these assumptions [24]. Ex vivo experiments on small samples of bone material excised from the human femur and tested to failure provide evidence for its brittleness at small strains [39]. Based on the above assumptions, the constitutive relationship

$$\sigma = H : \boldsymbol{\varepsilon} \text{ for } \Phi(\boldsymbol{\varepsilon}, \gamma) < 0 \tag{1}$$

is used to describe the stress–strain relationship everywhere in $\Omega$. Here, the variables $\sigma$, $H$ and $\varepsilon$ denote, respectively, the Cauchy stress tensor, the elastic stiffness tensor and the infinitesimal strain tensor. The strain $\boldsymbol{\varepsilon} = \frac{1}{2} (\text{Grad } u + (\text{Grad } u)^{\text{T}})$, where Grad is the gradient operator in the reference configuration and $u$ is the displacement vector. The operator: denotes a double contracted tensor product. A strain-based failure criterion is defined by the failure surface $\Phi$,

which is expressed in terms of the known failure strain level $\gamma$ (constant). Based on experimental and computational evidence [40,41], the elasticity of the bone material at any location within $\Omega$ is assumed to depend on the local vBMD $\rho$

$$H = h(\rho) \tag{2}$$

where the functional form of $h(\cdot)$ is assumed to be known. It is assumed that the internal stress $\sigma$ arises purely from the exchange of mechanical energy between the femur and the rest of the body, but the contribution from inertial stresses can be neglected. The validity of the latter assumption in small animals is directly supported by experiments [42]; in humans, an order-of-magnitude analysis (not shown here) unambiguously leads to the same conclusion. Thus, mechanical equilibrium at any location inside $\Omega$ is expressed as [43,44]

$$\text{Div } \sigma = 0 \tag{3}$$

Femur strength under a side-fall configuration is determined as follows. The external boundary of the volume $\Omega$ is partitioned into three mutually exclusive and exhaustive boundary surfaces. These partitions correspond to surfaces on which, respectively, external tractions are applied ($\Gamma_\tau$), displacement constraints are applied ($\Gamma_u$) and neither tractions nor displacement constraints are applied ($\Gamma_0$). The boundary $\Gamma_\tau$ is identified with the surface of femur head, and the boundary $\Gamma_u$ comprises surfaces corresponding to the greater trochanter and the distal end of the femur. The rest of the boundary of the volume $\Omega$ is identified with $\Gamma_0$. The boundary conditions are expressed as

$$\sigma \cdot N = \tau \text{ on the traction boundary } \Gamma_\tau \tag{4a}$$

$$u = a \text{ on the displacement boundary } \Gamma_u \tag{4b}$$

$$\sigma \cdot N = 0 \text{ on the free boundary } \Gamma_0 \tag{4c}$$

where the local outward normal at any point of the boundary is denoted by $N$, and the applied surface tractions $\tau$ and displacements $a$ are functions of location $x$, known up to a multiplicative constant. Consider the force vector obtained by integrating the surface tractions on $\Gamma_\tau$. Femur strength $S$ in side-fall configuration is defined as the smallest magnitude of this force vector such that a principal strain at any point on the femur neck surface reaches the failure limit [31].

The exchange of mechanical and biochemical energies between the femur and the rest of the subject's body are considered to be too complex to be abstracted in a meaningful manner. However, it is assumed that the effect of these exchanges on the change in bone strength can be expressed as

$$S(T^*) = S(0) + \int_0^{T^*} s(t)\mathrm{d}t \tag{5}$$

where the functional form of $s(\cdot)$ is known.

The single-scale model description given above is achieved above by setting out the question to be answered, then detailing the phenomena that are of interest within the portion of reality under inquiry, and finally by imposing specific assumptions that help to translate these phenomena into mathematical equations such as Eqs (1)–(5). The description of the problem in the hypothetical scale is then complete since, at least in principle, Eqs (1)–(5) can be solved to obtain $S(t = T^*)$, given the femur boundary $\Gamma$, the functional forms of $\Phi$, $h$ and $s$, the

constant $\gamma$, the initial spatial distribution of $\rho$ everywhere within the femur volume $\Omega$, the geometries of the boundaries $\Gamma_\tau$, $\Gamma_u$ and $\Gamma_0$ and the distributions of $a$ and $\tau$. Yet, the range of distances and time spans of the hypothetical scale, as specified in the previous section, is very large. As such, there exists no instrumentation that can fully inform the model. Thus, in practice, given the current experimental and computational limitations, it is not possible to solve the model.

**2.3.3. Empirical evidence of scale-dependent features.** The typical spatial resolution of clinical qCT imaging [0.625 mm, see 31] allows the important variations in spatial distribution of vBMD $\rho$ and femur boundary $\Gamma$ to be captured satisfactorily (Fig 1). The field of view of a typical CT scanner [50 cm, see 45] is sufficient to image the typical adult human femur. These distance measures, along with the range of characteristic time spans over which impact forces and strains on the femur surface vary, when measured experimentally [27,28], provide the grain and extent for a real scale labelled S1.

*Ex vivo* measurement of femur strength under fall loading conditions provides an estimate of loss of bone strength with age [33] (see Fig 3 in that paper). Here, donor ages are typically identified to the nearest 1 year ($10^8$ s) and the range of donor ages typically span several decades. Strain gauge resolution is 0.002 m [46] and boundary conditions are applied over the full length of the femur. These distances and time spans provide the grain and extent for scale S2.

The two scales "S1" and "S2" are used to populate the hypothetical scale described up to the previous section. Thus, for prediction of the change in strength of the adult human femur over a 10-year period:

i. Scale S1, $l^*_1 = 10^{-4}$ m, $t^*_1 = 10^{-3}$ s, $L^*_1 = 10^0$ m, $T^*_1 = 10^0$ s

ii. Scale S2, $l^*_2 = 10^{-3}$ m, $t^*_2 = 10^8$ s, $L^*_2 = 10^0$ m, $T^*_2 = 10^9$ s

The corresponding scale separation map is shown in Fig 2A.

**2.3.4. Modelling assumptions revisited.** Next one needs to modify each abstraction of reality, on which the hypothetical scale model in §2.3.2 was based, such that it is appropriate for the real scales introduced in the last section. The modified assumptions are listed below.

Scale S1

i. The observation domain ranges between $10^{-4}$ m and $10^0$ m in length and $10^{-3}$ s and $10^0$ s in time;

ii. The femur volume occupies a connected region in space and comprises a single material phase;

iii. The maximum distance between any two points in this region is smaller than $10^0$ m;

iv. When loaded at rates corresponding to a side-fall impact, the material phase of the femur responds in a spatially heterogeneous, rate-independent, isotropic linear elastic manner with failure at small strains;

v. Characteristic time spans of variations in fall-induced strains and tractions on the femur surface that are of interest are $10^{-3}$ s or larger;

vi. The contribution of inertia to fall-induced strains and tractions is relatively negligible;

vii. Below the failure strain, the elasticity of the material phase depends on the local vBMD;

viii. Characteristic distances of variations in the boundary separating the femur from the rest of the body and in the heterogeneity of the vBMD within the femur volume that are of interest are $10^{-4}$ m or larger;

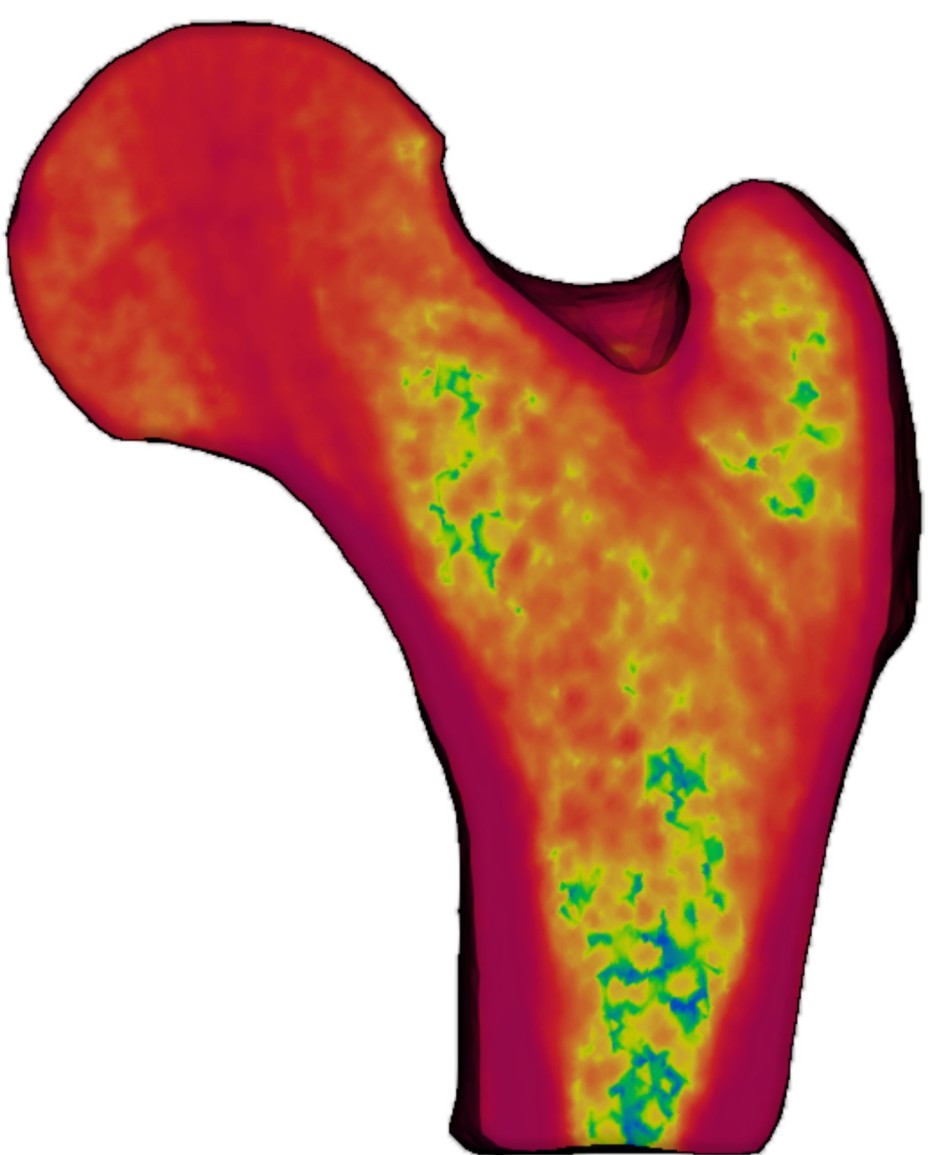

# vBMD (g/cm³) at age = 60 y

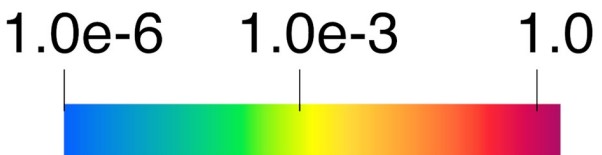

**Fig 1. Scale-dependent features informing the prediction of 10-year femur strength in a living human.** A cross-section of clinical quantitative computed tomography image set of an adult human femur at 0.625 mm resolution showing spatial heterogeneity in distribution of volumetric bone mineral density (vBMD). vBMD contour levels are shown in logarithmic scale to highlight regions of low vBMD. Details of cohort from which CT data is obtained and details of CT data processing are given in Ref. [31].

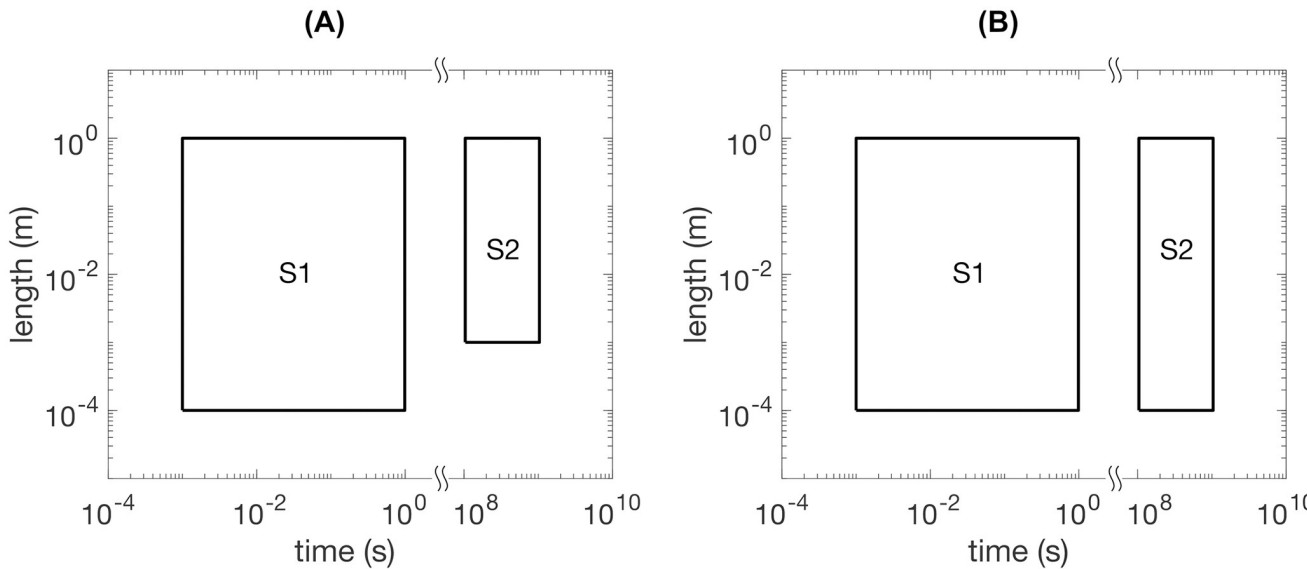

**Fig 2. Scale separation maps for predicting 10-year femur strength in a living human.** (A) Scale-dependent features identified using quantitative computed tomography (qCT) and ex vivo fall strength measurements. (B) Scale-dependent features identified using qCT only.

ix. Over durations that span at most $10^0$ s, the subject remains alive with sufficient metabolic rate such that mechanical and biochemical energy is exchanged continuously between the femur and the rest of the body;

x. Yet, such exchanges are so small over similar durations that the femur remains intact and net changes either in vBMD at any location or in femur strength or in femur boundary are all negligible.

Scale S2

i. The observation domain ranges between $10^{-3}$ m and $10^0$ m in length and $10^8$ s and $10^9$ s in time;

ii. Characteristic time spans of variations in bone strength that are of interest are larger than $10^8$ s;

iii. Temporal changes in bone strength that span longer than $10^9$ s are not of interest to the modelling problem;

iv. In ex vivo experiments used to determine femur strength, it is possible to capture the failure load to satisfactory precision by applying the displacement in increments larger than $10^{-3}$ m;

v. In the above experiments, the femur is constrained at locations separated by no more than $10^0$ m.

**2.3.5. Revised component hypomodels.** The revised set of assumptions above lead to the revision of the mathematical models at each scale (component hypomodels), along with redefinitions for each of the variables of the hypothetical scale model as follows.

Scale S1

All quantities that are meaningful in the domain of scale S1 are written with a superscript "[1]". For the purpose of determining femur strength $S^{[1]}$ over S1 time scales, the exchange of mechanical and biochemical energy between the femur and the rest of the body is neglected. Thus, the internal stress is meaningful only when a virtual load is applied in the fall configuration. The S1 scale model is given by the equations

$$\sigma^{[1]} = H^{[1]} : \varepsilon^{[1]} \text{ for } \Phi^{[1]}(\varepsilon^{[1]}, \gamma^{[1]}) < 0 \tag{S1 − 1}$$

$$H^{[1]} = h^{[1]}(\rho^{[1]}) \qquad\qquad \text{everywhere in } \Omega_{[1]} \tag{S1 − 2}$$

$$\text{Div } \sigma^{[1]} = 0 \tag{S1 − 3}$$

$$\sigma^{[1]} \cdot N^{[1]} = \tau^{[1]} \qquad\qquad \text{on the traction boundary } \Gamma_\tau^{[1]} \tag{S1 − 4a}$$

$$u^{[1]} = a^{[1]} \qquad\qquad \text{on the displacement boundary } \Gamma_u^{[1]} \tag{S1 − 4b}$$

$$\sigma^{[1]} \cdot N^{[1]} = 0 \qquad\qquad \text{on the free boundary } \Gamma_0^{[1]} \tag{S1 − 4c}$$

The model can be solved to obtain $S^{[1]}$ given the femur boundary $\Gamma^{[1]}$ $(= \Gamma_\tau^{[1]} \cup \Gamma_u^{[1]} \cup \Gamma_0^{[1]})$, the functional forms of $\Phi^{[1]}$ and $h^{[1]}$, the constant $\gamma^{[1]}$, the spatial distribution of $\rho^{[1]}$ within the femur volume $\Omega^{[1]}$, the geometries of the boundaries $\Gamma_\tau^{[1]}$, $\Gamma_u^{[1]}$ and $\Gamma_0^{[1]}$ and the distributions of $\tau^{[1]}$ and $a^{[1]}$. Note that for any given scale observations are always made over measurable and finite passages of time. Indeed, the grain and extent of measurement of time defines the scale itself. This is true for the scale S1. Yet, all quantities in the model (S1) presented above are independent of time. This is not an inconsistency at all. It is simply that the dependence of quantities on the passage of time has a negligible influence on determining femur strength.

Scale S2

All quantities that are meaningful in the domain of scale S2 are written with a superscript "[2]". The S2 scale model modifies the hypothetical scale model as follows. The exchange of mechanical and biochemical energies between the femur and the rest of the subject's body are considered to be too complex to be abstracted in a meaningful manner. Moreover, changes in femur strength $S^{[2]}$ over intervals smaller than $t^*{}_2$ are negligible. Thus, in order to determine the change in femur strength it suffices to consider only the *annual* rate of change $s^{[2]}(t^{[2]})$. In practice, femur strength can be measured experimentally only *ex vivo*. Hence, $s^{[2]}(t^{[2]})$ is obtained by regressing bone strength measurements from cadavers with $t^{[2]}$ which is identified with the donor's age at death. The S2 scale model is given by the equation

$$S^{[2]}(T_2^*) = S^{[2]}(t^{[2]} = 0) + \int_{t^{[2]}=0}^{T_2^*} s^{[2]}(t^{[2]})dt^{[2]} \tag{S2}$$

where the functional form of $s^{[2]}$ is known. The model can be then be solved given the initial value of strength $S^{[2]}(t^{[2]} = 0)$.

The multiscale problem requires models (S1) and (S2) to be solved together. The following are assumed to be known at the initial instant: the geometries of the femur boundary $\Gamma^{[1]}$ and its partitions $\Gamma_\tau^{[1]}$, $\Gamma_u^{[1]}$ and $\Gamma_0^{[1]}$ and the spatial distribution of $\rho^{[1]}$ within the femur volume $\Omega^{[1]}$. The functional form of $s^{[2]}(t^{[2]})$ is assumed to be known at all times. The distributions of $\tau^{[1]}$ and $a^{[1]}$, the functional forms of $\Phi^{[1]}$ and $h^{[1]}$ and the constant $\gamma^{[1]}$ are all time-independent and also assumed known. In order to close the multiscale problem, a relation model is needed to relate $S^{[1]}$ and $S^{[2]}$.

**2.3.6. Relation models.** Note that an infinite number of instants $t^{[1]}$ lie within the 1-year period identified by the age variable $t^{[2]}$. Without a model it is not possible to reconcile strength values defined at S1 and S2 scale time points. This is because of the so-called "curse of

resolution": on the one hand experimental measurement of femur strength is only possible *ex vivo*, while on the other repeated qCT scans are not possible due to risk of harm from radiation [47]. The relation model needed here is of homogenising type in time, which maps a first variable to a second where the first is defined at a higher resolution than the second. For simplicity, the following relation model is applied

$$S^{[2]}\left(t^{[2]}=0\right)=S^{[1]}\left(t_0^{[1]}\right) \tag{R12}$$

where $t_0^{[1]}$ is any instant within a $t^*_2$-sized interval around $t^{[2]}=0$. It is interesting to note that neither the equations comprising the hypothetical scale model Eqs (1)–(5) or the multiscale model (S1), (S2) and (R12) contain an expression for conservation of energy. This contrasts with the fact that exchange and balance of mechanical and biochemical energies were considered in the descriptions of the closed system. However, this is not altogether surprising, as it simply points to the fact that a statement of energy balance/exchange does not add any new knowledge about the system that is relevant to solve the problem at hand. Indeed, the same can be said about the balance of mass within the system. This is further underscored by the observation that the component hypomodels (S1) and (S2) and the relation model (R12) constitute a closed set of multiscale model equations describing the change in strength of an adult femur over a 10-year period. In contrast, conservation of momentum is important and appears as Eq (3) in the hypothetical scale model and as Eq (S1-3) in the S1 scale model.

**2.3.7. Orchestration.** Orchestration refers to the algorithmic aspect of solving the set of equations comprising the multiscale model. The orchestration for the simple multiscale model detailed above is shown in Fig 3A and comprises the following sequence of steps. First, model (S1) is solved to obtain femur strength at the initial time point. The initial time point corresponds to when the qCT scan of the femur is taken at a clinical facility. The qCT images provide the geometries of the femur boundary and its partitions and the spatial distribution of vBMD within the femur volume. The distributions of $\tau^{[1]}$ and $a^{[1]}$ corresponding to a fall loading situation, the functional forms of $\Phi^{[1]}$ and $h^{[1]}$ and the constant $\gamma^{[1]}$ are all empirically known. S1 scale strength is homogenised using model (R12) to obtain femur strength at scale S2, i.e. femur strength at age of clinical presentation. Last, model (S2) is executed to obtain femur strength at 10 years after clinical presentation, where the functional form of $s^{[2]}$ is given empirically.

**2.3.8. Verification of computability.** In the last step, the computability of the multiscale model as proposed is evaluated. We consider subject-specific data used in previous research to inform the multiscale model. In particular, we consider a subject from the study of Yang et al. [48]. The subject was a female from the Sheffield region and was 60 years old, healthy and had reached menopause at least 5 years prior, when the clinical CT evaluation was undertaken. Model (S1) detailed above is implemented as a finite-element analysis based on CT images. The analysis procedure, which includes specifications for $\tau^{[1]}$, $a^{[1]}$, $\Phi^{[1]}$, $h^{[1]}$ and $\gamma^{[1]}$, has been validated extensively in previous studies [30,31,41]. As such it can be used to predict the subject's femur strength at the time of the clinical CT evaluation. Model (R12) is a direct assignment operation, hence no further implementation is necessary to obtain the subject's femur strength at the age of 60 years, given the result of model (S1). For model (S2) it is assumed that the function $s^{[2]}\left(t^{[2]}\right)$ is a constant. The constant is the slope of linear regression of femur strength with age. Such a regression was obtained by Rezaei et al. [33] who conducted ex vivo experiments to measure side fall strength on femurs excised from cadavers of healthy elderly female donors whose age at the time of death was known. Thus, given the result of model (R12), i.e. subject's femur strength at the age of 60 years, executing model (S2) to obtain

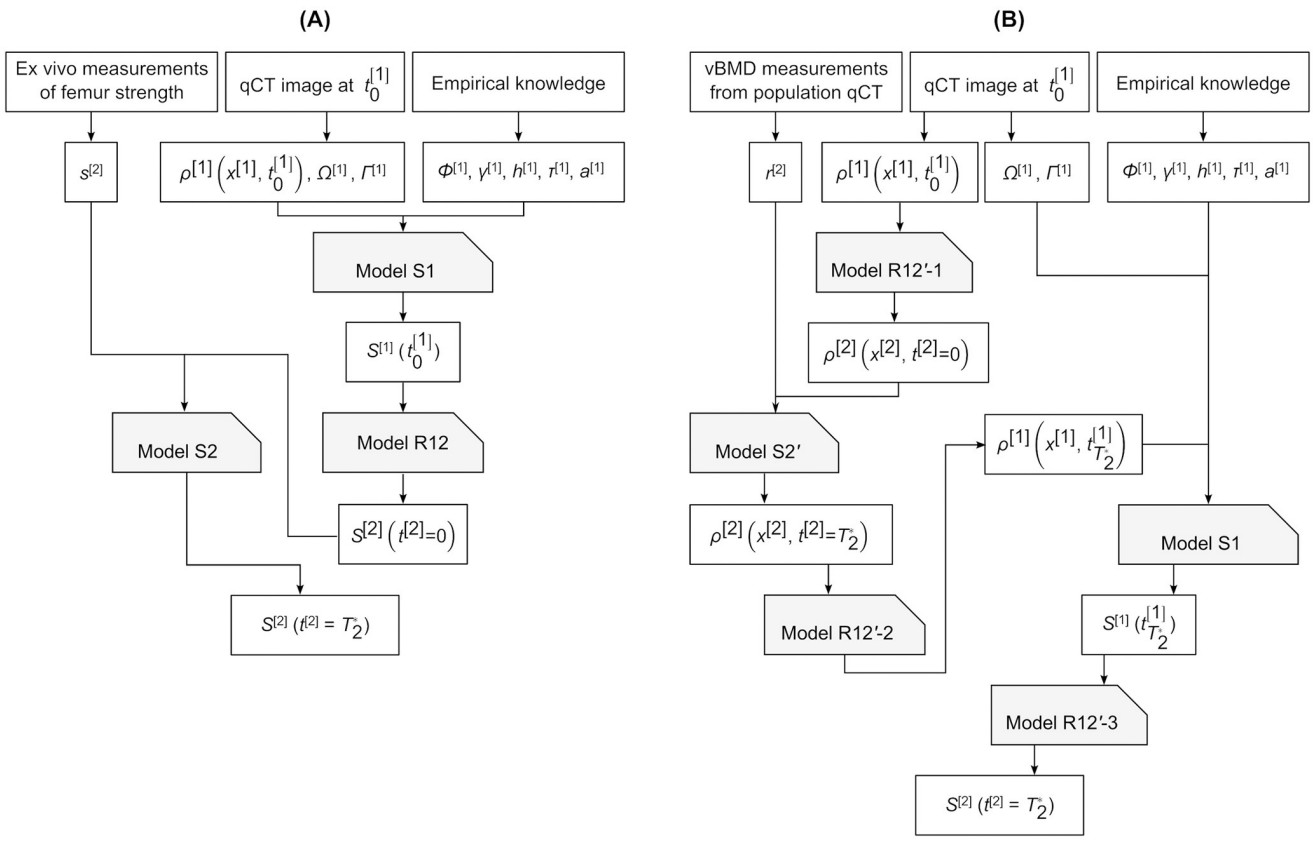

**Fig 3. Multiscale model orchestrations for predicting 10-year femur strength in a living human.** (A) Scale-dependent features identified using quantitative computed tomography (qCT) and ex vivo fall strength measurements. (B) Scale-dependent features identified using qCT only. Definitions of the mathematical symbols are given in the main text.

subject's femur strength at the age of 70 years is a simple algebraic operation. Altogether, the multiscale model is verified to be computable.

## 2.4. Dependence of multiscale model on instrumentation

This section investigates how the multiscale model illustrated above is modified when the set of experiments used to identify features of reality are changed. Refer back to the description in §2.3.1 where the evidence from ex vivo experiments regarding change of bone strength with age was employed. Consider replacing this observation with that bone loss occurs within the femur as a result of ageing. This is evident when qCT images are obtained in a population study where subjects vary by age, such as in [49] where vBMD averaged over trabecular regions of the whole femur was compared across subjects. Nevertheless, the domain of observation is the same as before, i.e. it ranges between $l^* = 10^{-4}$ m to $L^* = 10^0$ m in length, and from $t^* = 10^{-3}$ s to $T^* = 10^9$ s in time.

Following §2.3.2, the abstraction of the closed system is still based on the same assumptions that lead to Eqs (1)–(4). However, the assumptions that lead to Eq (5) need to be modified. In particular, it is assumed that the exchange of mechanical and biochemical energies between the femur and the rest of the subject's body are considered to be too complex to be abstracted in a meaningful manner. Yet, the effect of these exchanges on the change in vBMD can be

expressed as

$$\rho(x, T^*) = \rho(x, 0) + \int_0^{T^*} r(x, t) \mathrm{d}t \tag{5}$$

where the functional form of $r(\cdot)$ is known and $x$ denotes the position vector. Although the new hypothetical scale model can also be solved in theory, this is impossible in practice because of the large range of distances and times spans involved.

Following §2.3.3, scale dependent features are identified as before, except that ex vivo experiments are now replaced by population qCT datasets. Thus, the range of distances and times spans of scale S1 are unchanged, as these are informed by limitations of qCT evaluation of a single subject to predict bone strength. The range of distances and times spans of scale S2 are now informed by population qCT datasets. As before, information on subject age can be resolved up to the nearest 1 year ($10^8$ s) and typically spans several decades. As the longest feature of interest is 10-year change in vBMD, the temporal extent remains the same. The spatial grain and extent for scale S2 is the same as for qCT evaluation of a single subject. In summary:

i. Scale S1, $l^*{}_1 = 10^{-4}$ m, $t^*{}_1 = 10^{-3}$ s, $L^*{}_1 = 10^0$ m, $T^*{}_1 = 10^0$ s

ii. Scale S2, $l^*{}_2 = 10^{-4}$ m, $t^*{}_2 = 10^8$ s, $L^*{}_2 = 10^0$ m, $T^*{}_2 = 10^9$ s

The corresponding scale separation map is shown in Fig 2B.

Next, the model assumptions listed in §2.3.4 are modified, but these changes are limited to scale S2 only. These changes are as follows:

i. The observation domain ranges between $10^{-4}$ m and $10^0$ m in length;

ii. In population qCT datasets, it is possible to capture the change in vBMD to satisfactory precision by using an imaging resolution of $10^{-4}$ m and encompassing the femur region.

Referring to §2.3.5, it is noted that the S1 scale hypomodel remains unchanged, but the S2 scale component hypomodel needs to be modified according to the above changes. In order to determine the change in vBMD it suffices to consider only the *annual* rate of change $r^{[2]}(x^{[2]}, t^{[2]})$. As repeat qCT are impractical, determining this rate requires comparison of single time-point qCT images taken from subjects with varying ages. Hence, $r^{[2]}(x^{[2]}, t^{[2]})$ is obtained by regressing vBMD measurements from subjects with $t^{[2]}$ identified with subject's age at qCT evaluation and $x^{[2]}$ denoting an anatomically similar location in the femur across all subjects in the population. The S2 scale model is given by the equation

$$\rho^{[2]}(x^{[2]}, T_2^*) = \rho^{[2]}(x^{[2]}, t^{[2]} = 0) + \int_{t^{[2]}=0}^{T_2^*} r^{[2]}(x^{[2]}, t^{[2]}) \mathrm{d}t^{[2]} \tag{S2}$$

where the functional form of $r^{[2]}$ is known. The model can be then be solved given the initial vBMD distribution $\rho^{[2]}(x^{[2]}, t^{[2]}) = 0$. In order to solve the new multiscale model, the following are assumed to be known at the initial instant ($t^{[2]} = 0$): the geometries of the femur boundary $\Gamma^{[1]}$ and its partitions $\Gamma_\tau^{[1]}$, $\Gamma_u^{[1]}$ and $\Gamma_0^{[1]}$ and the spatial distribution of $\rho^{[1]}$ within the femur volume $\Omega^{[1]}$. The functional form of $r^{[2]}(x^{[2]}, t^{[2]})$ is assumed to be known at all times. The distributions of $\tau^{[1]}$ and $a^{[1]}$, the functional forms of $\Phi^{[1]}$ and $h^{[1]}$ and the constant $\gamma^{[1]}$ are all time-independent and also assumed known.

Yet, in order to couple the models (S1) and (S2′), a relation model is needed. For this, an approach similar to that in §2.3.6 is taken but the relation model now comprises

$$\rho^{[2]}(x^{[2]},\ t^{[2]} = 0) = \rho^{[1]}(x^{[1]},\ t_0^{[1]}) \qquad (R12' - 1)$$

$$\rho^{[1]}(x^{[1]},\ t_{T_2^*}^{[1]}) = \rho^{[2]}(x^{[2]},\ t^{[2]} = T_2^*) \quad (R12' - 2)$$

$$S^{[2]}\left(t^{[2]} = T_2^*\right) = S^{[1]}\left(t_{T_2^*}^{[1]}\right) \qquad\qquad (R12' - 3)$$

where $t_0^{[1]}$ and $t_{T_2^*}^{[1]}$ denote (as indicated by their superscripts) S1 scale instants, which are situated within (as indicated by the subscripts) $t^*_2$-sized intervals around the S2 scale instants $t^{[2]} = 0$ and $t^{[2]} = T_2^*$ respectively. The Eqs (R12′-1) and (R12′-3) are of homogenisation type in time, but Eq (R12′-2), where the order of mapping is reversed, is of particularisation type in time.

The model Eqs (S1), (S2′) and (R12′) are now closed but the orchestration requires substantial modification from that in §2.3.7. The sequence of model executions is: (R12′-1), (S2′), (R12′-2), (S1) and finally (R12′-3) (Fig 3B). These executions produce the following outputs respectively: vBMD distribution at the subject age when the qCT scan of the femur is taken at a clinical facility; vBMD distribution in the femur 10 years after this age; vBMD distribution at some (S1 scale) instant within the 10$^{\text{th}}$ year; femur strength at the above instant; and finally, femur strength at an age 10 years following clinical presentation. Note that the assumptions made earlier allow the geometries of the femur boundary and its partitions obtained from qCT images at instant $t_0^{[1]}$ to be used as is when model (S1) is solved at the instant $t_{T_2^*}^{[1]}$. The functional forms of $r^{[2]}$, $\Phi^{[1]}$ and $h^{[1]}$, the distributions of $\tau^{[1]}$ and $a^{[1]}$ corresponding to a fall loading situation and the constant $\gamma^{[1]}$ are assumed to be known empirically.

Lastly, referring to §2.3.8, the computability of the new multiscale model is verified with attention only to the hypomodel (S2′). This is justified as the hypomodel (S1) is unchanged and the relation model (R12′) comprises simple assignment operations. It is assumed that $r^{[2]}(x^{[2]}, t^{[2]})$ takes a constant value both in space and time. This constant value can be determined as the slope of a linear regression between vBMD averaged over the entire proximal femur and subject age, both taken from population qCT data. The technology to segment qCT images to obtain volume meshes already exists and is in fact a part of the implementation of hypomodel (S1). Qasim et al. [30] created vBMD-mapped volume meshes from qCT data of subjects taken from the study of Yang et al. [48], the same dataset from which the 60-year old subject mentioned above is drawn. Computing the volume averaged vBMD and its regression with age, given the vBMD-mapped volume mesh, is a straightforward operation.

Supporting Information §S2 details the same sequence of multiscale modelling steps as illustrated by the two examples above. However, the details of the resulting multiscale model differ because a large range of available instrumentations are included to describe a larger portion of reality relevant to the same problem (femur strength after 10 years in a live human). The portion of reality described is more reflective of the state-of-the-art of scientific knowledge in this field. Due to the complexity of phenomena occurring in this larger window of reality, a range of techniques from mathematical, physical and biological modelling are needed to mathematically describe the model. Supporting Information §S3 details the application of the multiscale modelling approach to a similar problem: femur strength after 8 weeks in a mouse. Here too, a large portion of reality is described that is representative of the state-of-the-art.

## 3. Results

Solving the multiscale model illustrated in §2.3 for the 60 year old female subject yields: using model (S1), femur strength for a neutral impact orientation at the time of CT evaluation to be

3719 N; using model (R12), femur strength at the age of 60 years to be 3719 N; using model (S2), and considering $s^{[2]}(t^{[2]}) = -85.9$ N/year [33], femur strength at the age of 70 years to be 2860 N, which is also the multiscale model prediction.

Regressing vBMD averaged over proximal femur volumes with subject age for $n = 96$ subjects in [48] gives the empirical rate of $r^{[2]}(x^{[2]}, t^{[2]}) = -0.00162$ g/cm$^3$/year. Solving the modified multiscale model illustrated in §2.4 for the same 60 year old female subject yields the following results. Using models (R12′-1), (S2′) and (R12′-2) in sequence, a modified vBMD distribution at an instant when the subject is 70 years old is obtained. Values of vBMD smaller than the $10^{-6}$ g/cm$^3$ threshold are set to the threshold value. The modified vBMD distribution is shown in Fig 4; restricting the lower limit of vBMD to be positive definite ensures a physically meaningful elastic response but does not significantly modify the femur strength prediction [30]. Using models (S1) and finally (R12′-3), femur strength for a neutral impact orientation at the age of 70 years to be 3487 N, which is also the modified multiscale model prediction.

Fig 5 shows the scale separation map and orchestration for the multiscale model representing the state-of-the-art in predicting femur strength after 10 years in a live human. Fig 6 shows the same for the multiscale model for prediction of femur strength after 8 weeks in the mouse.

## 4. Discussion

The objective of this study was to develop a systematic approach to scale separation based on the concepts of "grain" and "extent". We critically reviewed different methods used in the literature to decompose into a multiscale model a problem that cannot be solved using a single-scale model. In biomedical applications in particular, the most common approach seems to rely on the implicit assumption that there is a "natural" scale separation for the human body, and that this scale separation is provided by descriptive anatomy (body, organ, tissue, cell, organelles, etc.). But this makes little sense if we consider the shortest organ (pineal gland) is 6 mm, and the longest (sartorius muscle) is 600 mm. Additionally, anatomy does not provide any sensible temporal scale separation.

The necessity for multiscale modelling emerges from the limited spatiotemporal resolution and extent of the instrumentation used to observe the phenomenon of interest, and/or by the size/duration of the smallest and largest features of interest in that phenomenon. Thus, we propose that these two sets of information should define the separation of a problem across space-time scales. There might be application areas other than biomedicine where such necessity arises, hence we propose below a conceptually simple multiscale modelling process that is agnostic to the application as far as possible:

i. Postulating infinite resolution, formulate the mathematical model of a phenomenon with all its supporting idealisations (*hypothetical scale model*);

ii. Define the grain and extent of the different instrumentations used to observe the phenomenon, and of the smallest and largest features we want to capture in our model of the phenomenon;

iii. Derive from these an appropriate scale separation, represented with a *scale map*;

iv. Revise the idealisations used to build the hypothetical scale model for each separate scale;

v. Transform the infinite-resolution model into a collection of single-scale *component* hypomodels, each describing the phenomenon at the separate scales;

vi. Define the *relation* models required to transform quantities across different scales;

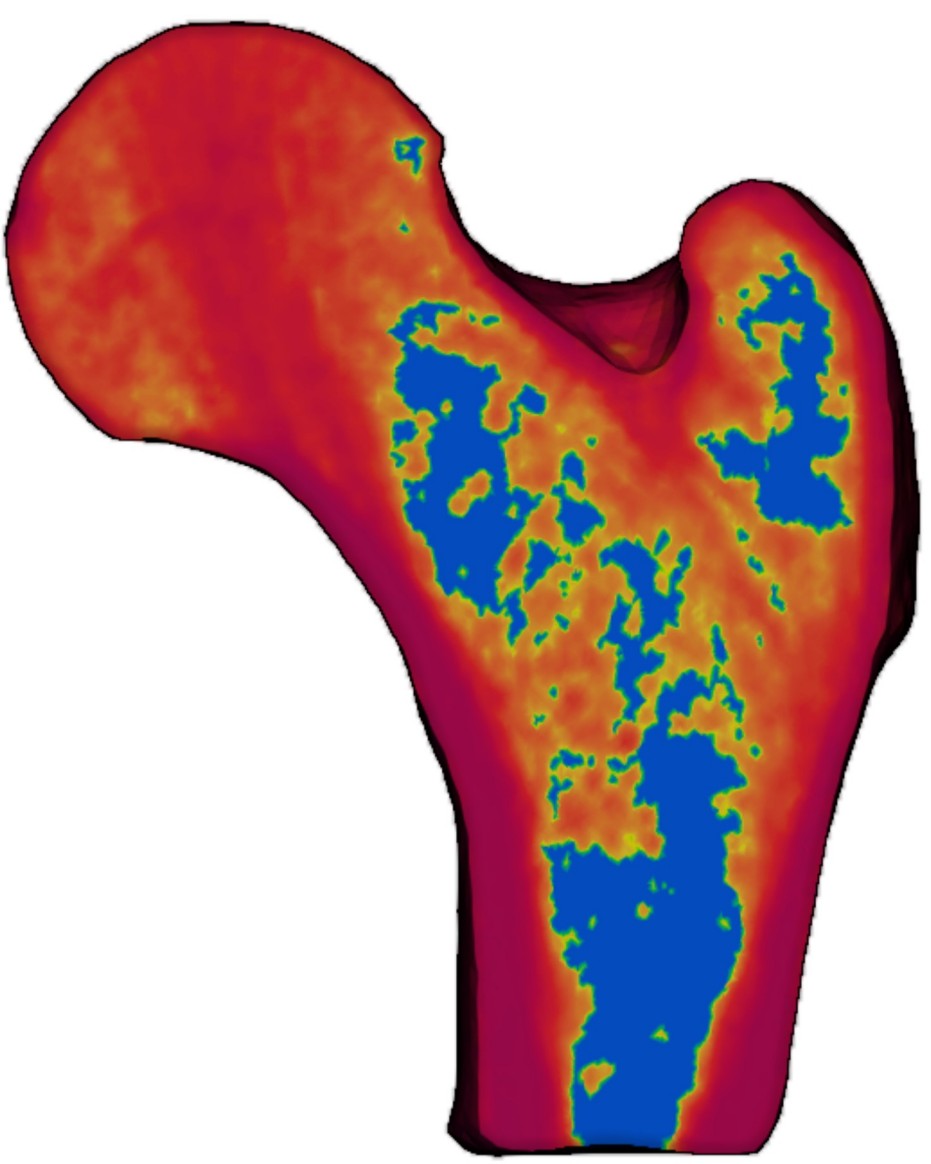

**vBMD (g/cm³) at age = 70 y**

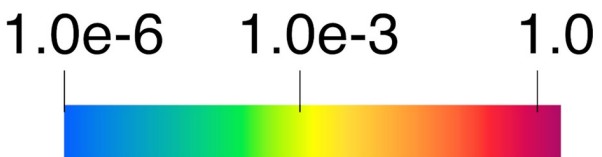

**Fig 4. Predicted volumetric bone mineral density (vBMD) distribution at a cross-section of an adult human femur.** The cross-section is identical to that in Fig 1 but after the application of models (R12′-1), (S2′) and (R12′-2) in sequence, resulting in a modified spatial distribution in vBMD at an instant when the subject is 70 years old. vBMD contour levels are shown in logarithmic scale to highlight regions of low vBMD, where the percentage change is highest.

**(A)**

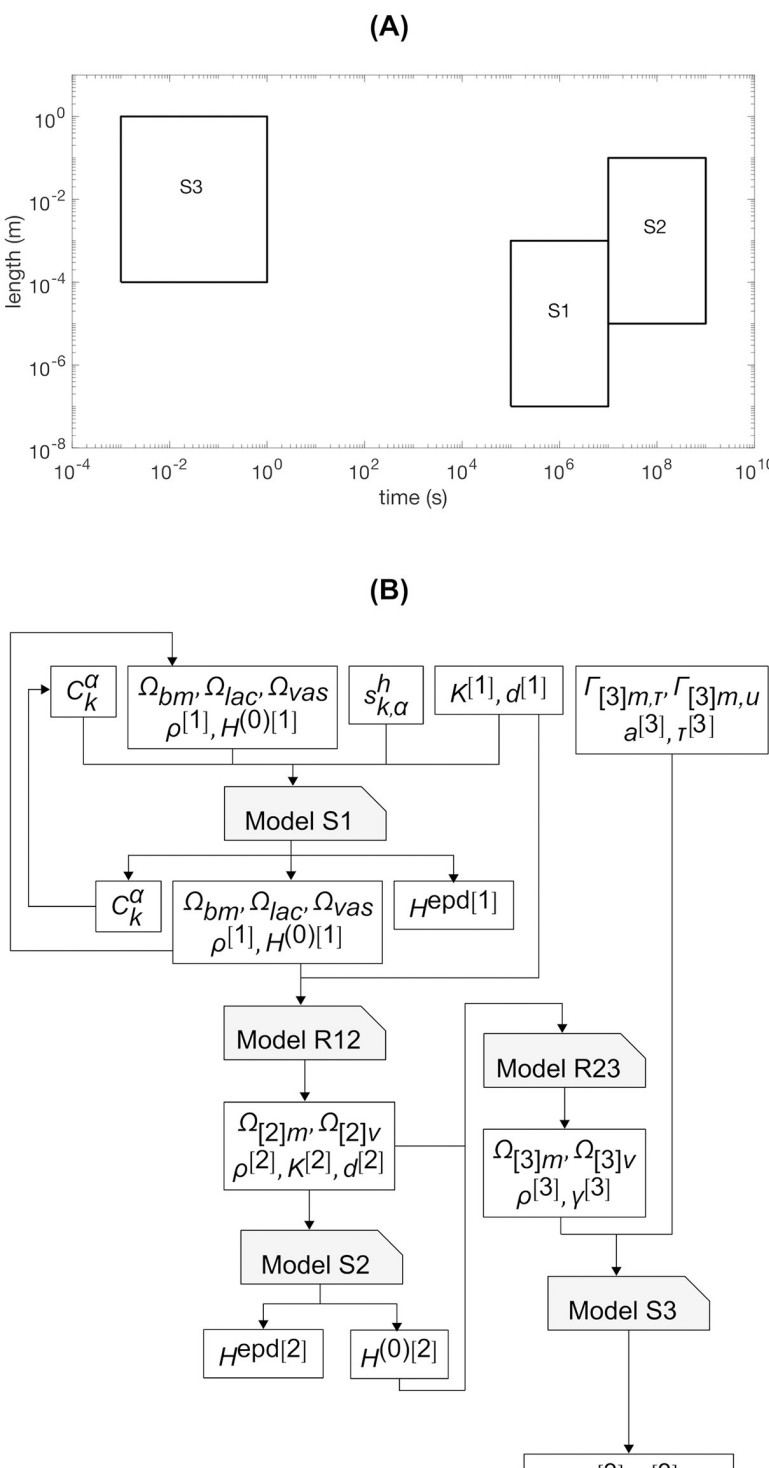

**(B)**

**Fig 5. Scale-separation map and orchestration for alternative multiscale model.** (A) Scale-separation map and (B) multiscale model orchestration. Several instrumentation capabilities are employed in this alternative model to identify scale-dependent features in a large portion of reality. The closed-loop workflow around scale S1 in the multiscale model orchestration indicates an update following a time increment.

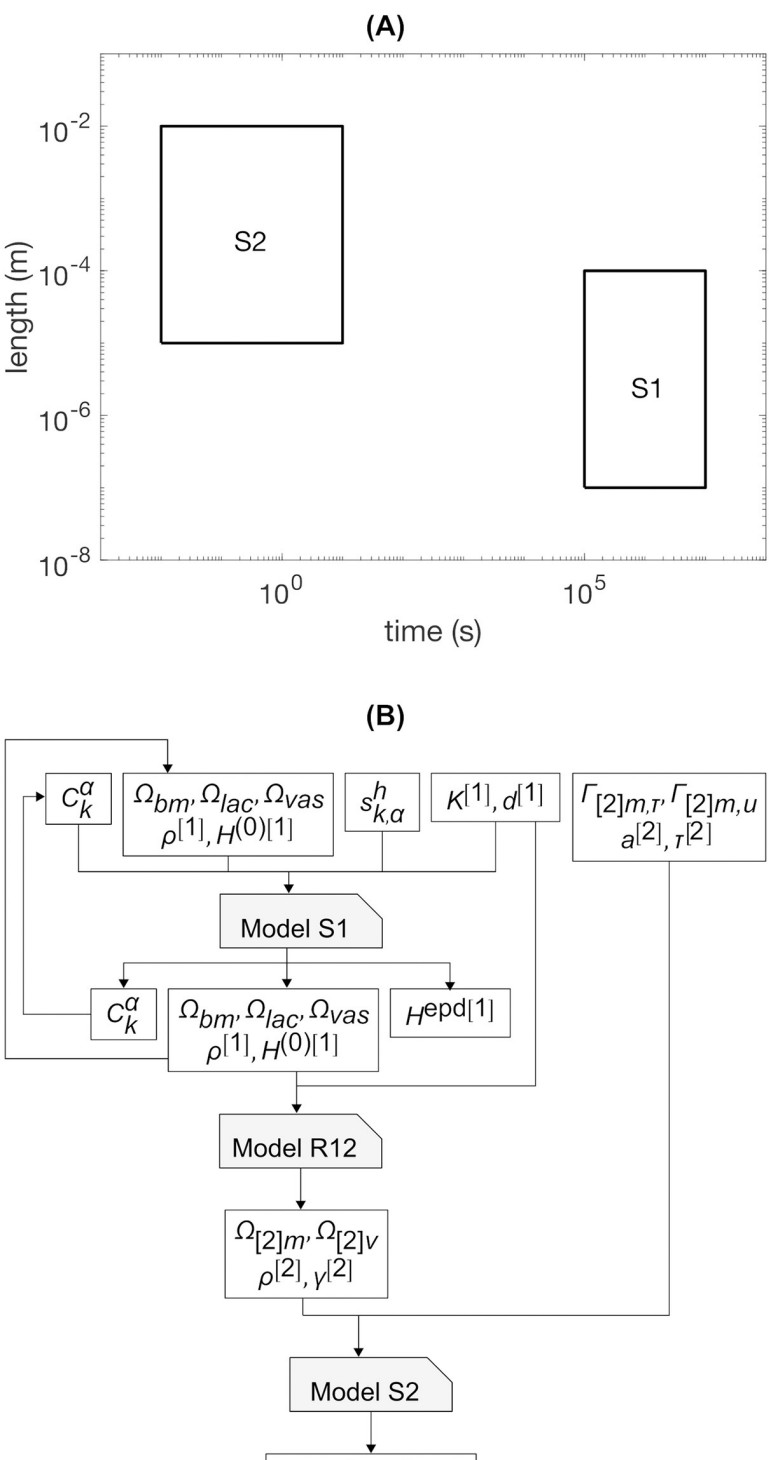

**Fig 6. Scale-separation map and orchestration for multiscale model to predict 8-week femur strength in a mouse.**
(A) Scale-separation map and (B) multiscale model orchestration. Instrumentation capabilities employed are appropriate for the small size of the animal. The closed-loop workflow around scale S1 in the multiscale model orchestration indicates an update following a time increment.

vii. Define the orchestration of the component and relation models into the multiscale model.

viii. Verify the *computability* of each component and relation model; if one model is too large to be computed with the available computational resources, it might be necessary to revise the scale mapping.

It should be noted that the proposed approach is not substantially different from what is done *in practice* in a number of multiscale modelling studies in biomechanics [see recent review in 50]. However, in most of these studies such a process remains implicit. Thus, some critical elements of the process, such as the hypothetical scale model or the criteria on which the scale separation is defined, are almost never provided. Note the explicit requirement of making the infinite resolution assumption in the proposed approach. This assumption implies that there is no feature so small in length or fast in time that it cannot be observed. Therefore, the only consideration that needs to be made in deciding whether to enter a feature into the model idealisations and equations, is whether it is of interest (step (i) above). The infinite resolution assumption forces the model (and the modeller) to distinguish features that are of interest from features that cannot be observed. The latter are accounted for further along the modelling process (step (iv)), and only after the instrumentations used to define real scales are formally recognised (steps (ii) and (iii)). Without these clarifications, critically reviewing and ensuring the reproducibility of multiscale models remains a challenge. In particular, not explicitly stating the experimental or computational basis for defining scale-dependent features of the problem might lead to incorrect comparisons between two multiscale models. Labelling scales as 'tissue' or 'organ' scale only makes this problem worse.

The credibility of any computational model is an important issue that needs to be considered for any new modelling approach [50–56]. Verification, validation, uncertainty quantification are the main aspects of model credibility that need to be assessed. Verification is mainly concerned with software encoding of the mathematical model and aspects of discretisation when continuous variables are used. Recall that the modelling process proposed in the present study requires the multiscale model computability to be evaluated (step (viii)). As soon as this is done, standard methods of model verification can be applied.

The level of validation needed depends on the specific question the model is targeted to answer [50,51,56], which requires the Context of Use (CoU) of the multiscale model to be specified. Recall that each real scale is identified with an experimental set-up. Hence, component hypomodels can be readily validated against experimental observations at the corresponding real scales. The CoU of the multiscale model will usually be different compared to the CoUs of its component hypomodels. Differences in CoUs reduce the credibility of each component hypomodel (by an unknowable degree) when used in the multiscale context. As an example, consider briefly the verification and validation of the S1-scale component hypomodel of the original multiscale model illustrated in §2.3. This model uses the FE method for predicting femur strength, a use that has been described extensively in past studies [24,30,31,57–60]. Here, a quadratic tetrahedral mesh with an average element size of 3 mm is used, which is sufficient to predict local strains on the femur neck with an accuracy of ~7% compared to strain gauge measurements in *ex vivo* studies. Based on a strain-based failure criterion, these strains are used to estimate the femur strength, the accuracy of which has been shown to be ~15% in an *ex vivo* setting. This level of accuracy is at par with other state-of-the-art qCT-based FE models of femur side-fall strength [24].

The validation of a relation model proceeds as follows. The relation model bridges two real scales and their corresponding component hypomodels (considered already validated). The smallest portion of reality that bounds these two real scales corresponds to a hypothetical scale (Fig 7). This portion of reality is carefully isolated such that following the same multiscale

modelling process described above, the resulting multiscale model comprises exactly the relation model under consideration and the two component hypomodels bridged by it and nothing else. This need not readily be the case, especially if the relation model and the component hypomodels were previously derived for a 'larger' hypothetical scale (and its corresponding 'larger' multiscale model) that comprises other real scales. Indeed, known effects from such other real scales should, as far as possible, be controlled for when isolating the 'smaller' hypothetical scale. The isolated portion of reality is then probed through separate experiments at the two real scales bridged by the relation model under consideration. Comparison between experimental observations and the smaller multiscale model prediction validates the relation model. A particular advantage available when designing these experiments is that any variable that appears both as the input of the relation model and as the output of a component hypomodel or vice versa need not be measured. However, the effect of isolating the smaller hypothetical scale will modify its CoU. This will reduce, again by an unknowable degree, the credibility of the relation model validated at the smaller hypothetical scale when it is used in the larger hypothetical scale context.

The modeller might seek to validate combinations of (individually validated) relation models by considering various 'small' hypothetical scales within the 'largest' (all inclusive) hypothetical scale. Each validation confirms the empirical understanding of the largest multiscale model and improves its credibility. Note that no hypothetical scale model (including the largest one) can ever be directly validated. Hence, validation of relation models (and their combinations) adds new knowledge and are necessary to ensure that the multiscale model gives the "right answers for the right reasons" [61].

Step (vii) of the modelling process proposed here details the orchestration of the multiscale model. This allows standard methods to be applied to quantify the uncertainty of the multiscale model or of any of its parts [52].

Another particular feature of the multiscale modelling approach presented here is that it results in no obvious ordering of scales, e.g. from 'finer' to 'coarser'. As indeed the illustrative examples show, one scale might be simultaneously finer in time relative to another scale (time extent of the first being smaller than the time grain of the second) and coarser in length (length grain of the first being larger than the length extent of the second). This leads to the need for relation models that are at once of the homogenisation type in time and of the particularisation (inverse of homogenisation) type in space. As mentioned before, validation of the relation models adds confidence in the empirical understanding of the system which can be considered as new knowledge.

It is interesting to compare the above features of the multiscale modelling approach presented here with those of AM&CM-based approaches. Fish [9] states that in AM&CM-based approaches, "A modelling and simulation approach is termed multiscale if it is capable of resolving certain quantities of interest with a significantly lower cost than solving the corresponding fine-scale system." Thus, the main motivation in an AM&CM-based approach is to reduce cost of computation, which contrasts with the current approach's motivation of understanding phenomena that cannot be observed using a single set of experiments. When using an AM&CM-based approach, one attempts to derive a "coarse-grained" model or a series of increasingly coarser-grained models (note the monotonic hierarchy) (Fig 7). This derivation employs 'transfer operators' (essentially, mathematical analysis techniques of varying sophistication), which ensures that all derived models are thermodynamically consistent with the 'fine-scale' model. To facilitate comparison, we correspond the hypothetical scale model of the present approach to the fine-scale model of the AM&CM-based approach, and somewhat more loosely, component hypomodels and relation models of the present approach to coarse-grained models and transfer operators of the AM&CM-based approach respectively.

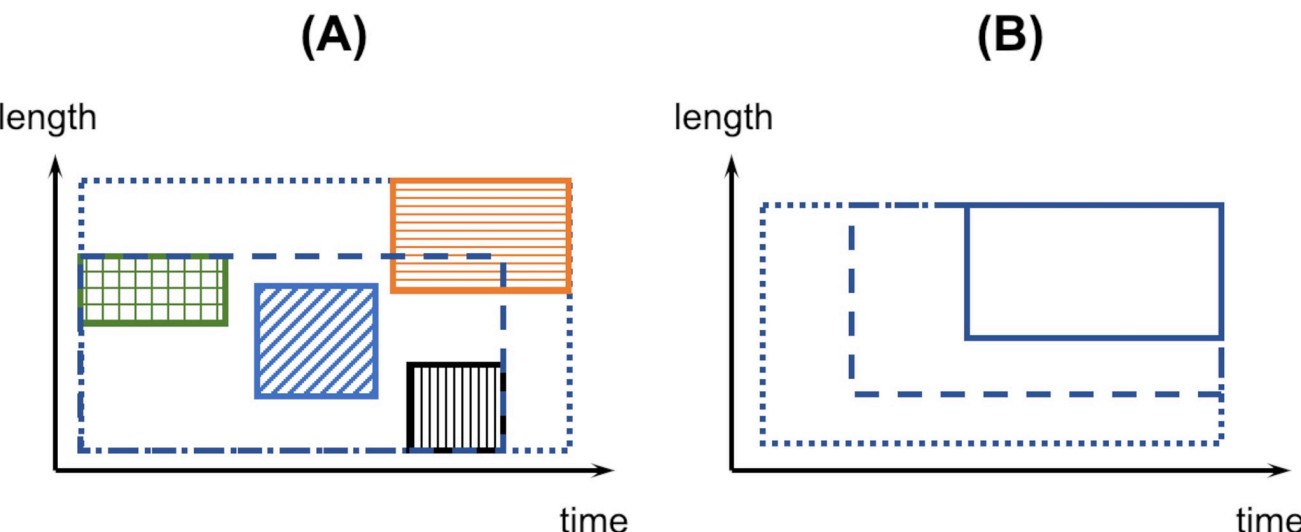

**Fig 7. Scale-separation maps for different multiscale modelling approaches.** (A) Scale-separation map for a multiscale model using the approach presented in this paper. In the absence of experimental capability to realise the real scale denoted by the box with a sloping stripe pattern, the relation model (not shown) bridging the real scales denoted by boxes with grid and vertical stripe patterns is validated by considering the isolated hypothetical scale shown by the dashed outline box. When isolating this hypothetical scale, known effects from other scales, such as that denoted by the horizontal stripe pattern, must be controlled for. When the real scale denoted by the sloping stripe pattern can be experimentally realised, more direct validation of this relation model is possible. (B) Scale separation map for an applied mathematics and continuum (micro)mechanics based multiscale model. Dotted outline box denotes the fine-scale or full model. The other two boxes denote successively coarser-grain models. The edges of each box denote the assumed range of validity of the corresponding model with no explicit link to experimental capability. "Coarsening" always leads to a model with a range of validity that is a subset of the range of validity of the original model.

Credibility assessment of an AM&CM-based model involves verification of mathematical and thermodynamic consistency of each transfer operator [10,13], which highlights the unique role the laws of conservation of energy and entropy play in AM&CM-based approaches. The fine-scale model and the transfer operators carry within themselves empirical knowledge of the system being modelled (e.g. organisation of material). Experimental validation of any of the coarser-grain models can confirm the empirical knowledge added by the transfer operator to the immediate (validated) finer-scale model. This assumes, rather excessively, that experimental capability is available to validate the finer-scale model. Note that the process of AM&CM based modelling is ambiguous to experimental set-ups to be used at each intermediate scale. Therefore, it is not clear whether in every instance empirical knowledge related to parts of the fine-scale system can be individually confirmed. In the absence of such a confirmation mechanism, 'fortuitously correct' results cannot be altogether excluded.

To illustrate the proposed methodological approach, we used as a guiding example a phenomenon of interest in our own research, in particular of determining the strength of a femur of a living human over a 10-year period, given a fixed loading orientation. Several assumptions were made in defining a closed system, and in defining relationships between quantities observable in the system. As expected, alternative abstractions of the same portion of reality led to the different multiscale models and model predictions. Specifically, the prediction of the original multiscale model (§2.3) is less individualised than that of the modified multiscale model (§2.4), as in the latter, the predicted change in femur strength depends on the subject-specific initial vBMD distribution. This individualised prediction of femur strength after 10 years is underscored by the individualised prediction of vBMD distribution after 10 years (Fig 4). Nevertheless, changing the set of abstractions did not necessitate a change in the general multiscale modelling approach, highlighting its versatility. Although the set of assumptions

made in §2.3 and §2.4 were based on scientific evidence, no attempt was made to fully capture the state-of-the-art of scientific understanding regarding the problem. The choice, motivated by brevity, is not a major limitation. Indeed, one may include a larger portion of reality, where "larger" could be in terms of range of distances and/or time spans, or in terms of quantities of interest. The two examples detailed in the Supporting Information section show that this only makes the individual steps of the modelling approach more complex, but not impossible. This increased complexity is emphasised in the scale separation maps and model orchestrations shown in Figs 5 and 6. A change in the modelling approach is not needed, which again underscores its versatility. This versatility is important because external factors (such as cost, feasibility, risk of harm from collecting information to feed the model and from making incorrect decisions based on model predictions) can require the modeller to choose a different set of abstractions or to consider a different portion of reality to be relevant.

It is a common observation that instrumentation capabilities continuously improve with time. Therefore, it is expected that for any problem the scale separation will evolve with time with the advancement in the state-of-the-art of experimentation and modelling. This could involve some qualitative changes. For example, two previously separated scales may merge into one. Another possibility is that new scales are added that stretch the range of distances and time spans of known reality. Yet another possibility is that new scales are added within the domain of known reality as a means to more directly validate relation models between two vastly separated scale models (Fig 7). Another possibility still is that new physical phenomena become of interest to the question being studied and thus warrant a place on the scale separation map. Given such a dynamically changing scenario, formally acknowledging the experiments on which the component scales of a multiscale problem are based should make the multiscale modelling process more rigorous.

As explained in the introduction, the problem addressed in this paper is a fairly neglected aspect of multiscale modelling, so there is very little literature to compare to. The only reference to limitation of measurement as a defining principle for scale, to the best of our knowledge, was made by Cushman [62]. However, that work was not concerned with a problem that possessed multiple scales, but rather with the problem of reconciling measurements performed at the scale of a laboratory experiment, and measurements performed at the scale of a field hydrology problem.

The main limitation of the proposed approach is that it is assumed that the phenomenon of interest can be described in a closed mathematical form under a set of acceptable idealisations and assuming infinite resolution of observations. We acknowledge that in some cases this is not possible, such as in systems where no reliable mechanistic knowledge is available and only statistical machine learning models are possible. In those cases, the process described here will need to be modified, although the key elements would probably remain unchanged.

## 5. Conclusion

In this study, we addressed the problem of formulating a multiscale model for phenomena where the quantities that need to be predicted and the quantities that are to be used for prediction occupy vastly different ranges of time and space in terms of their characteristic variation. The current literature on this problem does not provide a consistent definition of 'scale', the notion that should principally determine multiscale formulation. In particular, when solving a problem related to biomedical engineering, existing approaches for multiscale modelling drawn from a variety of fields of research appear inadequate. Thus, a definition of scale based on grain and extent was proposed, where grain and extent were determined based on the empirical evidence of limitations of existing measurement instruments or on smallest/fastest

and largest/slowest features of interest measured using these instruments. The new multiscale modelling approach based on this definition of scale was illustrated by considering a specific but well-researched problem. Here, we started by defining the quantities of interest to the problem and concluded with an algorithmic orchestration of component hypomodels and relation models, which when computed, led to a multiscale prediction. By varying either the instrumentation used to identify the features and scales of the problem, or the portion of reality included under investigation or the problem itself, the same approach resulted in three other model orchestrations, all of which differed qualitatively from the first. The first of these modified model orchestrations even resulted in a quantitatively different answer when executed. We conclude that the approach presented here is highly generalisable and can form a rational basis for formulating a wide range of future multiscale models.

## Supporting information

**S1A Fig.**
(TIF)

**S1B Fig.**
(PNG)

**S1 File.**
(DOCX)

## Author Contributions

**Conceptualization:** Pinaki Bhattacharya, Marco Viceconti.

**Formal analysis:** Pinaki Bhattacharya, Qiao Li, Marco Viceconti.

**Investigation:** Pinaki Bhattacharya, Qiao Li, Marco Viceconti.

**Methodology:** Pinaki Bhattacharya, Damien Lacroix, Visakan Kadirkamanathan, Marco Viceconti.

**Supervision:** Marco Viceconti.

**Writing – original draft:** Pinaki Bhattacharya, Marco Viceconti.

**Writing – review & editing:** Pinaki Bhattacharya, Damien Lacroix, Visakan Kadirkamanathan, Marco Viceconti.

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
