## [Decision Letter · Decision Letter 0]

9 Jul 2020

PONE-D-20-14710

A systematic approach to the scale separation problem in the development of multiscale models

PLOS ONE

Dear Dr. Bhattacharya,

Thank you for submitting your manuscript to PLOS ONE. After careful consideration, we feel that it has merit but does not fully meet PLOS ONE’s publication criteria as it currently stands. Therefore, we invite you to submit a revised version of the manuscript that addresses the points raised during the review process.

As you will notice, three reviews were received and they vary widely in their assessments. Reviewer 1 asked for only a couple minor revisions.  Reviewers 2 and 3 shared the opinion that the manuscript did not acknowledge a "large" body of literature in biomechanics and mechanics of materials, where scale separation has been addressed. These two reviewers also raised a number of questions/concerns regards the development and presentation of the models in the manuscript. 

We look forward to receiving your revised manuscript.

Kind regards,

Ryan K. Roeder, PhD

Academic Editor

PLOS ONE

Journal Requirements:

Reviewers' comments:

Reviewer's Responses to Questions

**Comments to the Author**

1. Is the manuscript technically sound, and do the data support the conclusions?

Reviewer #1: Yes

Reviewer #2: Partly

Reviewer #3: No

2. Has the statistical analysis been performed appropriately and rigorously? 

Reviewer #1: N/A

Reviewer #2: N/A

Reviewer #3: No

3. Have the authors made all data underlying the findings in their manuscript fully available?

Reviewer #1: No

Reviewer #2: Yes

Reviewer #3: No

4. Is the manuscript presented in an intelligible fashion and written in standard English?

Reviewer #1: Yes

Reviewer #2: Yes

Reviewer #3: No

5. Review Comments to the Author

Reviewer #1: The authors present a very nice systematic approach to multiscale modeling and apply this approach to the prediction of the fracture strength of a femur. The formulation of the model is explained clearly in the paper. The reviewer agrees with the authors that the details scale separation are often lacking in many publications and the method applied in this paper provides a methodology to better describe scale separation. In the reviewer's opinion, the manuscript is free of errors and is of very high quality. The reviewer feels that the manuscript is suitable for publication, upon addressing the following minor comments.

Minor Comments:

1) One item that seems to be missing is Table 1. There is a reference to Table 1 on line 279, but I was unable to locate Table 1 in the manuscript.

2) The The authors have done a extensive job of explaining the model setup and the execution of the model for the examples provided. However, the reviewer would like the authors to discuss in the manuscript the relevance of the changes in vBMD for the predicted in Figure 4. Without this type of physical interpretation of the results, it is difficult to assess the relevance of its predictive capacity.

Reviewer #2: The authors describe several aspects of multiscale modeling, with particular emphasis on bone biomechanics.

In the introduction, they deplore a situation where “most multiscale models in biomedicine“ would define scales in the sense of anatomical concepts, such as “body“, “organ“, “tissue“, “cell“ etc.

Quite strangely, they fully ignore the rich literature on continuum micromechanics, see e.g. the review in J Eng Mech 128, 808-816, 2002; and its applications in biomechanics, in particular bone biomechanics (see e.g. J Theor Biol 244, 597-620, 2007; Acta Mech 213, 131-143, 2010; and references therein), where the scale separation problem has been actually largely solved:

Instead of introducing anatomical concepts, it has turned out benedifical introduce so-called material volumes or representative volume elments (RVEs): For most of the practically encountered stiffness contrasts and volume fractions encountered in the context of bone biomechanics, the latter need to be separated from each other by a factor between 2 and 3, as has been shown both theoretically in a landmark paper of Drugan and Willis, J Mech Phys Solids 44, 497-524, 1999, and through experimental validations in the context of many applications on bone materials; see e.g. Eur J Mech 23A, 7893-810, 2004; Biomech Model Mechanobiol 2, 219-238, 2004; Transp Por Med 58, 243-268, 2005; J Theor Biol 260, 230-252, 2009; Ultrasonics 54, 1251-1269, 2014; Comp Meth Biomech Biomed Eng 17, 48-63, 2014; Int J Plast 91, 238-267, 2017. On the other hand, the largest RVE needs to be separated from the structural scale by a factor of 5 to 10, as e.g. shown experimentally, see Eng Str 47, 115-133, 2013.

Hence, the state of the art is far less deplorable than the authors state – and the actual question which arises is, how the results given by the authors would actually enrich the current state of the art.

In this context, a few critical issues arise, which need to be carefully discussed by the authors:

• In Section 2.1, the authors somehow compare the characterstic size of an organ L* with the “ignorable distance“ l*, which is reminiscent of the aforementioned length of the representative volume element – later on, the authors prefer the notion “grain“ in that context. The comparison between L* and l* looks plausible, but ignores an important aspect: It is not so much the overall length of the organ which needs to be separated from l*, but the characteristic length of the structural loading, norm of macroscopic stress over norm of gradient of macroscopic stress, i.e. a “wavelength“, see e.g. J Eng Mech 128, 808-816, 2002; Auriault et al, Homogenization of Coupled Phenomena in Heterogeneous Media, Wiley 2010.

• Model S1 seems to be related to a grain size of 0.1 mm; while model S2 seems to relate to a grain size of 1 mm – however, the manuscript is not fully clear about the second choice; as on page 18 and 19, sizes of 0.1 mm are given with respect to l2* are given (these are probably missprints). In this context, it is important to repeat that RVE-lengths (or grain sizes) need to be separated in scale, from heterogeneities inside the RVE: In this context, the choise of 0.1 mm does not make sense: this is the size where either vascular pores or extravascular bone matrix can be observed, but no “continuous bone phase“. Hence, from a rigorous continuum micromechanics viewpoint, model S1 is invalid. On the other hand, as the authors write themselves, S2 is the biomechanical “standard model“ – and albeit not referenced by the authors, such models have also been developed in a full multiscale manner, with clearly defined “grain sizes“ and “grain constituents“, see e.g. Ann Biomed Eng 36, 108-122, 2008; Int J Num Meth Biomed Eng 32, e02760, 2016; for CT-related application, and Comp Meth Appl Mech Eng 254, 181-196, 2013; Bone 64, 303-313, 2014; Bone 107, 208-221, 2018; for the problem of bone remodeling.

• Concerning the last statement of the authors’ abstract, they may be reminded that multiscale mechanics models have been transforming various engineering fields, such as concrete or timber engineering, see e.g. Cem Concr Res 34, 67-80, 2004; Eur J Mech 24A, 1030-1053, 2005; Int J Eng Sci 147, 103196, 2020, and references therein.

Reviewer #3: PONE-D-20-14710

The manuscript lacks in many aspects. The multiscale modeling is a rich subject in applied mathematics and continuum mechanics. The manuscript is without necessary understanding of the state-of-the-art in multiscale modeling, continuum mechanics as well as in the computational science and engineering. The article lacks necessary mathematical rigor. The article is not recommended for publication.

Please see for example:

J. Fish, Bridging the scales in nano engineering and science, J. Nanopart. Res. 8 (2006) 577–594.

M.G.D. Geers, V.G. Kouznetsova, W.A.M. Brekelmans, Multi-scale computational homogenization: trends & challenges, J. Comput. Appl. Math. 234 (2010) 2175–2182.

Technical comments:

1) The scale separation is a mathematical construct that can be use only in certain case. Spatial and temporal scales can only be separated if the error from the asymptotic expansion is sufficiently small. Many multiscale problems cannot be asymptotically expanded. Please see Principles of Multiscale Modeling, Weinan E, Cambridge University Press, 2011.

2) The manuscript talks about mechanical and biochemical energies yet it fails to introduce the energy equation. The manuscript lacks necessary understanding of conservation laws of mass, momentum and energy as well as of the second law of thermodynamic.

3) The coupling of scales usually requires carefully crafter transfer operators. In mechanics of materials, this relation is known as the macro-homogeneity condition. This is important for the model consistency. Every multiscale model has to converge (e.g., in the weak sense) to the full model. This is clearly not guaranteed in this work.

4) The introduction of functions S is very ad hoc and does not follow from any balance law, an averaging principle nor from thermodynamics.

5) The model talk about time, yet the balance law in Eq. (3) is quasi-static one. Therefore, taking about spatial and temporal scales is dubious. There can be perhaps some pseudo time, but that is poorly described. Multiscale modeling of temporal scales is very much open scientific problem and this manuscript is lacking any understanding.

6) The main manuscript body and the supplementary material are not consistent. The main manuscript uses the small strain formulation yet the supplementary material talks about finite strains.

7) Modeling of damage is complex process. It is well known that computational model, if not properly derived, will lose ellipticity and lead to erroneous results. How do you prevent the artificial localization? Please see Energy-Based Coupled Elastoplastic Damage Models at Finite Strains, J. W. Ju, Journal of Engineering Mechanics, 1989.

8) The equation (2) in the supplementary material is the rate depended equation. How do you integrate this and assure the frame invariance? This equation is for the second Piola-Kirchhoff stress tensor, but the evolution equations are usually written in the current configuration using the Cauchy stress. How do you obtain elastoplastic-damage modulus tensor? You need the pull-back operations. Again, the lack of mathematical rigor is vast.

9) The manuscript lacks any concepts of verification and validation in computational science end engineering. Please see Patrick J. Roache, Verification and validation in computational science and engineering, Hermosa, 1998.

10) What is the numerical method used? If the finite element method, what type of element is used? How if the model discretized? Was solution verification performed?

6. PLOS authors have the option to publish the peer review history of their article (what does this mean?). If published, this will include your full peer review and any attached files.

Reviewer #1: No

Reviewer #2: No

Reviewer #3: No

---

## [Author Response · Author response to Decision Letter 0]

11 Jan 2021

Responses to Reviewer #1 comments 

Comment R1.1

The authors present a very nice systematic approach to multiscale modeling and apply this approach to the prediction of the fracture strength of a femur. The formulation of the model is explained clearly in the paper. The reviewer agrees with the authors that the details scale separation is often lacking in many publications and the method applied in this paper provides a methodology to better describe scale separation. In the reviewer's opinion, the manuscript is free of errors and is of very high quality. The reviewer feels that the manuscript is suitable for publication, upon addressing the following minor comments.

Response

We appreciate the reviewer for their comment.

Comment R1.2

Minor Comments:

1) One item that seems to be missing is Table 1. There is a reference to Table 1 on line 279, but I was unable to locate Table 1 in the manuscript.

Response

We thank the reviewer for pointing out this omission. The information intended to be presented in the missing table was already presented in the text of §2.3.3. Hence, for the sake of brevity, the reference to the missing Table 1 has been removed at line 301 in the revised manuscript.

Comment R1.3

2) The authors have done an extensive job of explaining the model setup and the execution of the model for the examples provided. However, the reviewer would like the authors to discuss in the manuscript the relevance of the changes in vBMD for the predicted in Figure 4. Without this type of physical interpretation of the results, it is difficult to assess the relevance of its predictive capacity.

Response

Both the original multiscale model (Fig 3A) and the modified multiscale model (Fig 3B) predict a change in femur strength over 10 years. Both models start with the same information at the initial time point i.e. quantitative computed tomography (qCT) image data. In the original multiscale model, it is assumed that the change in femur strength over a 10-year period is a universal value and the function "s" ^"[2]" in model S2 is taken from a population-based regression [24]. In the modified multiscale model, it is assumed that the rate of change in femur strength is not universal, and in particular, is dependent on the subject’s femur geometry and the initial vBMD distribution. Indeed, the predicted femur strength after 10 years is different for the two models.

The relevance of Fig 4 is that it explains the difference between the two predictions. In the modified multiscale model, the change in vBMD per unit time is a universal (not subject-specific) and homogeneous value; it is given by the function "r" ^"[2]" in model S2� and derived from population-based regression of data obtained in Yang et al. [1]. Yet, this universal and homogeneous change is superposed (as shown in Fig 4) on the initial distribution of vBMD over the femur geometry. As both initial vBMD distribution and femur geometry are subject-specific, the predicted change in femur strength is also subject-specific. Lines 739–744 in the revised manuscript highlight this relevance.

 

Responses to Reviewer #2 comments

Comment R2.1

a) The authors describe several aspects of multiscale modeling, with particular emphasis on bone biomechanics.

b) In the introduction, they deplore a situation where “most multiscale models in biomedicine” would define scales in the sense of anatomical concepts, such as “body”, “organ”, “tissue”, “cell” etc.

c) Quite strangely, they fully ignore the rich literature on continuum micromechanics, see e.g. the review in J Eng Mech 128, 808-816, 2002; and its applications in biomechanics, in particular bone biomechanics (see e.g. J Theor Biol 244, 597-620, 2007; Acta Mech 213, 131-143, 2010; and references therein), where the scale separation problem has been actually largely solved.

Response

a) Our examples are drawn from bone biomechanics because of our familiarity with this area of research. The approach to multiscale modelling developed in the present study is not intended to be limited to this application. This point was already made in the previous version (see lines 127¬–129 in the revision) but is now further emphasised in the revised manuscript at lines 110–111 and 590–592.

b) In paragraphs 3–6 of the Introduction section, various existing multiscale modelling approaches were presented. The intent was to highlight the differences in the underlying motivations between these approaches and the approach presented here. In our view, continuum micromechanics approaches fall under the type of approach described in paragraph 4 of the Introduction. In response to the reviewer’s comment, this view is clarified in the revised manuscript (lines 72–75). Where we talk about “most multiscale models in biomedicine” we are not referring specifically to continuum micromechanics approaches. This is also clarified in the revised manuscript (lines 89–91).

Continuum micromechanics based multiscale modelling approaches are a subset of applied mathematics and continuum mechanics (AM&CM for brevity) based approaches, which are also referred to extensively in the comments of Reviewer #3. Hence, in the following, we use “AM&CM” to denote continuum micromechanics as well.

c) It is not possible to present an exhaustive review of any of these approaches, as each is a vast field in itself. Such an undertaking would better suit a review article, which the present study is not. In response to the reviewer’s comment, the revised manuscript now includes references to reviews of application of AM&CM to biomechanics and to bone biomechanics in particular (line 79). It is clarified that these references should not be construed as exhaustive (line 102).

We agree with the reviewer in that the AM&CM approach has successfully solved several multiscale problems in biomechanics, and especially in bone biomechanics. Yet, our view is that several multiscale problems in biomechanics, including bone biomechanics, are far from being solved. A specific example of such a problem is that considered in §2.3 of the manuscript, which to the best of our knowledge has not been investigated using an AM&CM approach. Please also see the response to comment R2.2 below. More generally, the terms “multiscale modelling” and “scale separation” can imply different notions depending on which approach is used; indeed, this is the main message of the Introduction section. Hence, we respectfully differ from the claim that the scale separation problem in bone biomechanics has been largely solved.

Comment R2.2

Instead of introducing anatomical concepts, it has turned out beneficial to introduce so-called material volumes or representative volume elements (RVEs): For most of the practically encountered stiffness contrasts and volume fractions encountered in the context of bone biomechanics, the latter need to be separated from each other by a factor between 2 and 3, as has been shown both theoretically in a landmark paper of Drugan and Willis, J Mech Phys Solids 44, 497-524, 1999, and through experimental validations in the context of many applications on bone materials; see e.g. Eur J Mech 23A, 783-810, 2004; Biomech Model Mechanobiol 2, 219-238, 2004; Transp Por Med 58, 243-268, 2005; J Theor Biol 260, 230-252, 2009; Ultrasonics 54, 1251-1269, 2014; Comp Meth Biomech Biomed Eng 17, 48-63, 2014; Int J Plast 91, 238-267, 2017. On the other hand, the largest RVE needs to be separated from the structural scale by a factor of 5 to 10, as e.g. shown experimentally, see Eng Str 47, 115-133, 2013.

Response

We agree with the reviewer’s comment. In AM&CM based multiscale models of bone, an RVE-based definition of “scale” has yielded useful results. We thank the reviewer for the excellent reference to Drugan [2] which shows that even when scales are separated by a finite factor, one may expect predictions of reasonable accuracy. These advances are briefly reviewed in the revised manuscript (lines 76–80).

As mentioned in the response to comment R2.1, not all multiscale problems in biomechanics have been solved. One specific example is the problem considered in §2.3 of the manuscript. It is not clear how successful an AM&CM approach will be for this problem, or more generally, for all multiscale problems in biomechanics. In the following, the studies referenced by the reviewer are briefly considered with this question in mind.

Hellmich et al. [3] estimated the “ultrastructural” stiffness Cultra (1–10 μm “scale”) in the human femur. One can imagine further homogenising Cultra to obtain stiffness at the “microscale” [defined as 100 μm to several mm in 3]. In turn, the microscale stiffness can be used to predict femur strength using methods described in detail elsewhere [4]. Hellmich et al. [3] validated the predicted ultrastructural moduli of human femur against elastic moduli that were either inferred from sonic velocities measured on 2 mm specimens [5] or measured directly on 5 mm specimens [6]. In both cases, the experimental specimens were excised from human femora. We ignore here the discrepancy between the scales compared by Hellmich et al. [3]: 1–10 μm in the model vs 2 mm / 5 mm in the experiments. Even so, their lowest prediction error (model II, longitudinal Young’s modulus) is ~1.11 GPa. This is ~ 2.5x higher than that of the regression model of Morgan et al. [7] which has a standard error of 443 MPa. In the single scale model (model S1), quantitative computed tomography (qCT) data is used as input to Morgan et al. [7] model to obtain heterogeneous bone elastic properties and then on to predict femur strength. A larger prediction error implies poor site-specificity, and the error of 1.1 GPa is comparable to the 1.5–4.5 GPa range of variation of elastic moduli in the femur neck [7]. It is well known that the accuracy of elastic properties is the most important factor determining the accuracy of ex vivo femur strength prediction [8, 9]. Hence, it is doubtful that if the model of Hellmich et al. [3] was used to predict femur strength, the resulting accuracy would be higher than that of the S1 scale model of §2.3. 

The absence of a “microscale” stiffness prediction in Hellmich et al. [3] was resolved in Hellmich et al. [10]. Although “microscale” RVEs were redefined as having dimensions of 5–10 mm, the scale discrepancy between model and experiments was removed. Unfortunately, the prediction error of Hellmich et al. [10] model for human bones [11, 12] is 29–39%, which is still much larger than that of [7] which is ~ 15%.

Morgan et al. [7] and Taddei et al. [9] concerned prediction of bone strength, which can only be measured ex vivo. In the in vivo setting, multiple studies [13-16] have shown that the regression model of Morgan et al. [7] coupled with an FE modelling approach based on Taddei et al. [9] predicts bone strength values that can classify fracture status very accurately. It is not clear whether the model of Hellmich et al. [10] gives similar accuracy in vivo. The assurance of in vivo accuracy is essential to solve the example multiscale problem considered in §2.3.

The models of Hellmich and co-workers [17-21] predict plasticity, viscoelasticity and damage responses of bone at the microscale. As such, these models require inputs at various underlying “scales” in order to characterise the hierarchical organisation of bone. In the above studies, these inputs were obtained from data reported in literature on ex vivo samples. Typical inputs needed were volume fractions of spaces occupied by vascular (in “bone microstructure”) and lacunar (“extravascular bone material”) material. It remains unclear whether such data be obtained in an in vivo setting as well, with sufficient precision to detect site- and subject-specific variability. Without such assurance, it is doubtful that ex vivo accuracy can be replicated.

Lastly, the scale separation in time is not accounted for in any of the AM&CM-based approaches discussed above. This demonstrates that the success of AM&CM approaches is partial, and there is still a need for developing a new approach. A more general point is that every model is useful only for answering a specific question. This is also true for any multiscale model. While the multiscale models referenced by the reviewer are extremely useful to understand how the hierarchical organisation of bone controls its mechanics in general, they are not very useful in answering the multiscale question considered in §2.3 where subject-specificity and in vivo capability are important considerations.

Comment R2.3

Hence, the state of the art is far less deplorable than the authors state – and the actual question which arises is, how the results given by the authors would actually enrich the current state of the art.

Response

The main result of the present study is the novel approach of defining scales and of constructing a multiscale model. The prediction of 10-year femur strength is an illustration of this main result. Therefore, in the following, we only discuss three added benefits of this new approach.

The proposed approach allows to communicate the process of multiscale model creation in a systematic and organised fashion (a sequence of 8 clearly distinct steps). To the best of our knowledge, this sequence of steps has not been clearly spelled out elsewhere, although some of these are often employed by modellers. The approach emphasises model creation steps that are beyond the description of the mathematical machinery employed.

The organisation of the modelling process is rooted in the language of experimental evidence. This has two benefits. The first is that it is not necessary to assume the validity of a hypothetical scale model (or “fine-scale” model in the AM&CM sense). This assumption is the basis of all AM&CM approaches but does not yield useful results in all contexts of use.

The second benefit is that for a model created using this approach, it should be straightforward to determine exactly which aspect(s) of the model should be revised following an improvement in experimental capability.

These advantages are now clarified in the Discussion section of the revised manuscript, which has been significantly expanded.

Comment R2.4

In this context, a few critical issues arise, which need to be carefully discussed by the authors:

 In Section 2.1, the authors somehow compare the characteristic size of an organ L* with the “ignorable distance” l*, which is reminiscent of the aforementioned length of the representative volume element – later on, the authors prefer the notion “grain” in that context. The comparison between L* and l* looks plausible, but ignores an important aspect: It is not so much the overall length of the organ which needs to be separated from l*, but the characteristic length of the structural loading, norm of macroscopic stress over norm of gradient of macroscopic stress, i.e. a “wavelength“, see e.g. J Eng Mech 128, 808-816, 2002; Auriault et al, Homogenization of Coupled Phenomena in Heterogeneous Media, Wiley 2010.

Response

We thank the reviewer for their comment. We agree with the reviewer that in the continuum mechanics approach, one needs to derive a “macroscale” formulation that is consistent with a “microscale” formulation in the limit of asymptotic “scale” separation. (The quotes emphasise that these words have specific meanings in that approach.) The microscale equations involve mechanical variables. Hence, separation of length scales is defined in terms of lengths over which mechanical variables characteristically vary, giving rise to notions of macro- / micro- stresses.

However, the approach presented in this manuscript should in no way be confused with the AM&CM approach (see response to comments R2.1–2.3). Specifically, here l* is not the dimension of a representative volume element (RVE) in the sense of an AM&CM approach. Similarly, the reviewer’s comment regarding the size of L* is relevant when using an AM&CM approach, but not in the present approach.

In the present approach, separation of scales simply means that there is no single set of instruments that can characterise the portion of reality that is being modelled (i.e. hypothetical scale). The dimensions l* and L* are simply the bounds of the hypothetical scale. The hypothetical scale is decomposed into real scales, but there is no underlying hierarchy such as “micro”-“meso”-“macro” between these scales (unlike in AM&CM). Component models at each real scale are formulated based on what is observable at that scale. The credibility of the component models is assured by experiments performed at the corresponding scale. Therefore, there is no need to separately ensure that component models at different scales are asymptotically equivalent to each other or to the hypothetical scale model. This is clarified in lines 635–639 and lines 697–705 of the revised manuscript.

Comment R2.5

 Model S1 seems to be related to a grain size of 0.1 mm; while model S2 seems to relate to a grain size of 1 mm – however, the manuscript is not fully clear about the second choice; as on page 18 and 19, sizes of 0.1 mm are given with respect to l2* are given (these are probably misprints). In this context, it is important to repeat that RVE-lengths (or grain sizes) need to be separated in scale, from heterogeneities inside the RVE: In this context, the choice of 0.1 mm does not make sense: this is the size where either vascular pores or extravascular bone matrix can be observed, but no “continuous bone phase”. Hence, from a rigorous continuum micromechanics viewpoint, model S1 is invalid.

Response

The multiscale modelling approach presented here is different from existing multiscale modelling approaches in several aspects. One key aspect is that a real (not hypothetical) scale is defined always based on the capabilities of experimental instrumentation being used. In particular, the instrumentation informs the grain and the extent of the corresponding real scale. Obviously, the same portion of reality can be observed using different sets of instrumentations. This will lead to different idealisations, different multiscale models and different predictions. This dependence of prediction on the choice of instrumentation set used in not an artificial one. In fact, the modeller faces such choices in everyday contexts. Moreover, each choice has cost, feasibility and ethical consequences which, independent of model predictive accuracy, can determine which model(s) remains usable in a given context. The revised text on lines 734–769 clarifies this in detail.

The manuscript attempted to illustrate this dependence on instrumentation set through the original multiscale model illustration (presented in §2.3) and its modified form (presented in §2.4). The grain and extent of scale S1 are identical for both multiscale models (Fig 2A and Fig 2B) because the same experiment is used to define it. Specifically, the grain in the length axis is determined by the resolution of clinical qCT (0.625 mm) which is used to construct FE models. After rounding down to the nearest power of 10, we get the grain as 10–4 m, as indicated in the main manuscript.

As the reviewer points out, S2 scale grains in the length axis are different: for the original multiscale model (Fig 2A) the S2 scale grain is 1 mm, whereas in the modified multiscale model the (Fig 2A) the S2 scale grain is 0.1 mm. The difference is because of difference in experiments used to define these scales. In the original multiscale model, S2 scale is based on ex vivo measurement of bone strength. Hence the grain of length is determined by the resolution of strain gauges (2 mm). The reviewer is gratefully acknowledged for pointing to the misprints; in the previous version of the manuscript the resolution incorrectly referred to displacement measurement instead of strain measurement and the value was incorrectly typed as 0.002 mm instead of 0.002 m (now corrected on lines 297–298 of the revised manuscript). In the modified multiscale model, S2 scale is based on measuring local changes in vBMD as obtained from clinical qCT. Hence the grain of length is determined by the resolution of clinical qCT (0.625 mm). After rounding down to the nearest powers of 10, we get the grains as 10–3 m and 10–4 m for scale S2 in the original and modified multiscale models respectively.

Interpreting these grain values as RVE dimensions is a misunderstanding. There is no relation between scales defined here and scales in the sense of AM&CM based approaches (please see response to earlier Comment R2.4).

Comment R2.6

 On the other hand, as the authors write themselves, S2 is the biomechanical “standard model” – and albeit not referenced by the authors, such models have also been developed in a full multiscale manner, with clearly defined “grain sizes” and “grain constituents”, see e.g. Ann Biomed Eng 36, 108-122, 2008; Int J Num Meth Biomed Eng 32, e02760, 2016; for CT-related application, and Comp Meth Appl Mech Eng 254, 181-196, 2013; Bone 64, 303-313, 2014; Bone 107, 208-221, 2018; for the problem of bone remodeling.

Response

Unfortunately, a careful search for the text “standard model” in both the Main Manuscript and the Supporting Information (SI) section did not lead to any successful hits. Hence, we are unable to respond to the reviewer’s comment in full.

Both the original and modified multiscale models incorporate the effects of bone remodelling, but do not include a mechanism for it. The multiscale models illustrated in the SI section include a mechanism for bone remodelling in humans and mice. This is done by considering a larger portion of reality to be relevant than that considered in the main text. Including a mechanism of bone remodelling allows its effects to be considered in a more individualised manner. For example, one may specify the parameter "f" ("s" _"k,α" ^"h" ) in Eq. (S1-iii) identically across all individuals in a population. Yet, other individual details e.g. the initial spatial distribution of "C" _"k" ^"α" will modulate the effect of this parameter, leading to an individualised prediction. Alternatively, or in addition, the function "f" ("s" _"k,α" ^"h" ) can be specified individually, leading to a further individualised prediction. The choice depends ultimately on what the modeller considers relevant and is able to measure.

The mechanism for bone remodelling considered in the SI section was purely biochemical in nature. Yet, as mentioned in SI §2.8, bone remodelling mechanisms involving mechanoregulation such as those considered in [22], [23] and [24] can be introduced using the same modelling approach. It only remains to specify a link between "s" _"k,α" ^"h" and σ[1]. As mentioned earlier, it is not our objective here to propose a specific mechanism for bone remodelling, or for bone strength over a long duration. Rather, the objective is to demonstrate that a complete system of multiscale model equations can be formulated and solved by the consistent application of a definition of scale that is based on experimental capabilities. This, and the abundance of mechanoregulation models of bone remodelling reported in the literature, discouraged us from citing a select few such models in this manuscript.

The models of Hellmich et al. [25] and Blanchard et al. [26] are similar to the S1 scale model presented in the main text, in that these are informed in part by clinical qCT. However, that is where the similarity ends. The models of Hellmich et al. [25] and Blanchard et al. [26] pertain to the mechanics of the human mandible and human vertebra respectively, unlike the mechanics of the whole human femur as considered in this manuscript. Our view is that models are not truths in themselves, but useful representations of truths. The usefulness of a model is ultimately linked to the truth it is supposed to represent. Missing an experimental validation, it is challenging to extrapolate the usefulness of the models of Hellmich et al. [25] and Blanchard et al. [26] to the application of prediction of 10-year strength of human femur.

Comment R2.7

 Concerning the last statement of the authors’ abstract, they may be reminded that multiscale mechanics models have been transforming various engineering fields, such as concrete or timber engineering, see e.g. Cem Concr Res 34, 67-80, 2004; Eur J Mech 24A, 1030-1053, 2005; Int J Eng Sci 147, 103196, 2020, and references therein.

Response

Several domains of engineering contain problems of a multiscale nature, where significant improvement in predictive accuracy is still needed. We believe that the multiscale modelling approach presented in this paper is well posed to meet this need across engineering domains, working both by itself and with existing multiscale modelling approaches. This was the intended import of the last statement of the abstract. We hope that our responses to the previous comments will satisfy the reviewer in accepting this import. 

Responses to Reviewer #3 comments

Comment R3.1

a) The manuscript lacks in many aspects. The multiscale modeling is a rich subject in applied mathematics and continuum mechanics.

b) The manuscript is without necessary understanding of the state-of-the-art in multiscale modeling, continuum mechanics as well as in the computational science and engineering. 

c) The article lacks necessary mathematical rigor.

The article is not recommended for publication.

Please see for example:

J. Fish, Bridging the scales in nano engineering and science, J. Nanopart. Res. 8 (2006) 577–594.

M.G.D. Geers, V.G. Kouznetsova, W.A.M. Brekelmans, Multi-scale computational homogenization: trends & challenges, J. Comput. Appl. Math. 234 (2010) 2175–2182.

Response

a) We thank the reviewer for their comments. It is acknowledged that multiscale modelling is a rich subject with contributions from several disciplines including applied mathematics and continuum mechanics (AM&CM for brevity). In the original version a very limited introduction to AM&CM based approach to multiscale modelling was provided (paragraph 4 of Introduction) in order that the manuscript length was reasonable. However, after reflecting on the reviewer’s comments, we accept that this brevity failed to clarify the distinction between AM&CM based approaches and the method proposed here. We have now addressed this lacuna in the Discussion section of the revised manuscript, which has been significantly expanded. In particular, lines 706–733 compare the two approaches head-to-head. This is only an exercise in clarification; a comparison of pros and cons in not meaningful as the two approaches potentially excel in different contexts of use. For a better understanding of the state-of-the-art in AM&CM based multiscale modelling we have cited the works suggested by the reviewer in addition to those mentioned already (lines 76–79 in the revised manuscript).

b) However, a fully developed understanding of AM&CM based multiscale modelling is unnecessary to appreciate the particulars of the proposed approach. Hence, no further changes have been made to demonstrate a full understanding.

c) The absence of mathematical rigour in the manuscript follows from the above. In an AM&CM based approach, scale is often a mathematical construct (see response to Comment R3.2). Thus, conducting an independent experiment at an “observational scale” is often not conceivable, and cannot be used for the purpose of obtaining the mathematical form of the coarse-grained model. The usual recourse is to derive this from the mathematical form of the finer-scale system. This then necessitates an application of mathematical rigour.

In contrast, for the type of multiscale problems that we are mainly concerned with, component models at each scale state the modeller’s hypotheses at that scale. The validity of the mathematical form of these models is given by experiments that define that particular scale (see response to Comment R3.10). Given this alternative, application of mathematical rigour in our approach is of very limited use than in AM&CM based approach.

Comment R3.2

a) Technical comments: 1) The scale separation is a mathematical construct that can be used only in certain case. 

b) Spatial and temporal scales can only be separated if the error from the asymptotic expansion is sufficiently small. Many multiscale problems cannot be asymptotically expanded. Please see Principles of Multiscale Modeling, Weinan E, Cambridge University Press, 2011.

Response

a) We request the reviewer to reconsider that “scale separation” can have different meanings in different disciplines and in different multiscale modelling approaches. In the Introduction section, four different meanings of the term were reviewed based on published literature. These second of these, referred to in paragraph 4 of the Introduction section, is that of a purely mathematical construct. However, this specific meaning was intended only in paragraph 4 of the Introduction. Everywhere else in the manuscript (including in the title) it has either the general meaning given in paragraph 2 of the Introduction or a different specific meaning which is clear from the context.

b) Our understanding is that the “mathematical construct” meaning is commonly used when one physical idealisation is valid over the entire space–time range. In such contexts, multiscale modelling can leverage some level of self-similarity or space–time periodicity present in the phenomenon of interest. For instance, in Bhattacharya et al. [27] “scale” is identified by the continuous-valued exponent of the local Reynolds number. This allows one to solve the NS equation, which is the one physical idealisation that is valid over the entire space–time range. Attaching a physical meaning to the exponent is unnecessary for the analysis carried out in Bhattacharya et al. [27]. In other words, it is purely a mathematical construct.

Even limiting to this “mathematical construct” meaning, we submit that all multiscale models can be asymptotically expanded. The real question is whether the prediction made by such an asymptotic expansion is useful or not. One can argue that such a model is useful if the error in the expansion is smaller than that of any other existing model. In other words, it does not have to be “sufficiently small” in a numerical sense; rather, it only has to be smaller than any competing model. In the field of bone biomechanics, scale separation by a factor between 2 and 3 has been shown to produce satisfactorily accurate results in some contexts [2] (see lines 76–78 in the revised manuscript).

Comment R3.3

2) The manuscript talks about mechanical and biochemical energies, yet it fails to introduce the energy equation. The manuscript lacks necessary understanding of conservation laws of mass, momentum and energy as well as of the second law of thermodynamic.

Response

References to exchange of mechanical and biochemical energies are made in the manuscript in §2.1, 2.3.1 and 2.3.2 leading up to the hypothetical scale model. These acknowledge the existence of the neighbourhood of the closed system being modelled. Knowledge of energy exchange with the neighbourhood is limited to the effect it has on properties of the system being modelled e.g. boundary tractions, internal stresses, bone strength, expressions for which have been introduced in the model. The remaining references to exchange of energy in the main text and SI section have similar meanings.

Regarding energy balance within the closed systems, an energy equation was not introduced as the systems of equations for each of the illustrated models were already closed. This usually implies, as is the case here, that a statement of energy balance within the system does not add any new knowledge about the system. The same can be said about the balance of mass within the system. In contrast, conservation of momentum was important in several closed systems illustrated in the manuscript. These are expressed in Eq (3) and (S1-3) in the main text and Eqs (1), (H-i), (S3-i) and (S2�-i) in the SI section. The modelling approach presented here does not preclude introducing balance equations of mass, momentum, energy or any other variable for a problem where such equations are required to close the system, or where these equations contain new knowledge. Lines 395–406 of the revised manuscript clarify this argument.

One application where conservation law of energy and the second law of thermodynamics provides new knowledge is the thermodynamically consistent construction of transfer operators in an AM&CM-based approach. Indeed, this is the focus of the studies of Fish [28] and Geers et al. [29] and many others. We have included these references where the interested reader can find guidance for using these laws in the context of AM&CM based methods. However, this application is not universally useful. In the approach presented here, the nearest equivalents of transfer operators are called relation models. As the Discussion section of the revised manuscript clarifies on p.27, relation models can be individually validated experimentally, thus obviating a need for demonstrating thermodynamic consistency.

Comment R3.4

3) The coupling of scales usually requires carefully crafted transfer operators. In mechanics of materials, this relation is known as the macro-homogeneity condition. This is important for the model consistency. Every multiscale model has to converge (e.g., in the weak sense) to the full model. This is clearly not guaranteed in this work.

Response

The reviewer’s remarks pertain to AM&CM based approaches where the challenge is to derive coarse-grain models that converge to the “full” model. The careful crafting of transfer operators is necessary because independent experimental validation of coarse-grain models at each scale (often merely a mathematical construct) is not routine practice.

As already discussed, this is not the only type of multiscale modelling problem/approach. The approach presented in this paper is applied for problems the full model is not widely accepted. Hence, experiments define scales, and observations from these experiments are employed to construct / validate component hypomodels for the corresponding scales. The component hypomodels at each scale are coupled using relation models. Unlike transfer operators in the AM&CM approach, which need to satisfy mathematical and thermodynamic consistency conditions, relation models in the present approach are simply statements of the modeller’s hypothesis. The validity of relation models is tested by combining experiments at multiple scales.

Comment R3.5

4) The introduction of functions S is very ad hoc and does not follow from any balance law, an averaging principle nor from thermodynamics.

Response

The mathematical form of function such as s in Eq. (5) express the modeller’s hypotheses of the system behaviour. Although determining these from laws of balance, averaging or thermodynamics is not precluded, it is equally justified to develop and validate these based on experiments.

Comment R3.6

a) 5) The model talks about time, yet the balance law in Eq. (3) is quasi-static one. Therefore, taking about spatial and temporal scales is dubious. There can be perhaps some pseudo time, but that is poorly described. 

b) Multiscale modeling of temporal scales is very much open scientific problem and this manuscript is lacking any understanding.

Response

a) In the following, the use of Eq (3) for representing the mechanics at scale S1 is justified. Note that Eq (3) predicts internal stresses induced due to a fall. Fall induces inertial and contact forces (contact with the ground). An order-of-magnitude analysis shows that the contribution to internal stresses due to inertial forces is relatively negligible. At scale S1, the orders of magnitude of volumetric bone mineral density, elastic modulus of bone and extent of femur are 1 g/cm3, 2 GPa and 1 m respectively. The failure strain in compression is known to be of the order of 1%. Hence, a minimum acceleration of the order of 2 x 104 m/s2 (= (2 GPa x 1%) / (1 m x 1 g/cm3)) is necessary in order that contributions from inertial forces and external surface tractions are similar. This is several orders of magnitude higher than accelerations typically induced in falls, which are of the order of 10 m/s2. Furthermore, experimental studies involving impact loading of cadaveric femurs show that femur strength depends very weakly on the loading rate and viscoelastic effects are negligible [30-33]. Hence, for predicting femur strength the time history of external surface tractions τ may be neglected. This is clarified on lines 227–228, 244–246 and 325 of the revised manuscript.

In conclusion, the quasistatic loading in Eq (3) is justified by experimental observations. It amounts to saying that the effect of passage of time on mechanical response at scale S1 is negligible. It is incorrect to infer from this equation that there is no passage of time at scale S1. We hope this clarifies that the notion of quasistatic loading and the specification of temporal grain and extent are compatible with each other. The clarification further supports the rationale for scale separation in time for the illustrated problem, which was already explained in the manuscript.

b) We fully accept the reviewer’s observation that using a quasistatic Eq. (3) in the illustrated example does not fully illustrate the wide applicability of our approach. However, a more sophisticated illustration is not necessary because the wide applicability is assured by a rather straightforward argument. The argument is that in the present approach component hypomodels are validated by experiments at corresponding scales (see response to Comment R3.10). This validation justifies the time-dependent form used in the component hypomodel equations. There is no additional requirement of ensuring that these mathematical forms are consistently derived between scales separated by time. This is different from AM&CM based approaches of driving scale separation of the time axis. We believe that the reviewer’s concern regarding the problem of scale separation using AM&CM approaches, which as explained earlier, is not relevant for the present approach.

Comment R3.7

6) The main manuscript body and the supplementary material are not consistent. The main manuscript uses the small strain formulation, yet the supplementary material talks about finite strains.

Response

The objective of the manuscript is to develop a multiscale modelling approach, not a specific multiscale model. The use of this approach is illustrated by developing several multiscale models for exemplar problems, but it is obviously not intended to be limited to these. Hence, it is not at all intended that multiscale models constructed using this approach should be consistent with each other with regard to mathematical formulation.

Indeed, the exemplar multiscale models presented in the main manuscript and the supplementary materials are distinct from each other. The distinction arises from the window of reality we seek to capture using the model, as is demonstrated by the different descriptions of the closed system. For example, §2.3.1 states:

“The femur volume is assumed to comprise a single material phase that fails in brittle fashion at small strains when loaded at physiological strain rates.”

In contrast, SI §S2.1 states:

“The bone region is assumed to comprise two material phases: a bone matrix phase and a non-bone matrix phase.”

Both descriptions relate to the same object, i.e. the adult human femur. Yet, these descriptions are different because in the first we are limited by the spatial resolution of clinical qCT imaging, while in the second we are limited by the spatial resolution of a microscope.

The difference in choice of instrument is not at all artificial and has significant implications on key aspects of the multiscale modelling. This is clarified better in the revised manuscript on lines 734–746.

Comment R3.8

7) Modeling of damage is complex process. It is well known that computational model, if not properly derived, will lose ellipticity and lead to erroneous results. How do you prevent the artificial localization? Please see Energy-Based Coupled Elastoplastic Damage Models at Finite Strains, J. W. Ju, Journal of Engineering Mechanics, 1989.

Response

We agree with the reviewer that modelling damage is a complex process, and that some derivations of the model might lead to results such as artificial localisation. We take the view that the “correctness” of models matters only as far as their “usefulness”. Thus, if a damage model predicts localised damage at multiple sites with some being artificial, and still predicts the experimentally observed bone strength better than any other model, then the modeller is justified in claiming the error to be acceptable and therefore to use the model.

The same notion can be applied to a very large body of models and complex processes. The paper details a multiscale modelling approach that can accept all these models. Therefore, specifying any one model in detail does not serve the purpose of illustration. Indeed, our objective was not to propose a damage model at all, but just to state that some formulations e.g. Ju [34] exist and can be applied.

Comment R3.9

a) 8) The equation (2) in the supplementary material is the rate depended equation. How do you integrate this and assure the frame invariance?

b) This equation is for the second Piola-Kirchhoff stress tensor, but the evolution equations are usually written in the current configuration using the Cauchy stress.

c) How do you obtain elastoplastic-damage modulus tensor? You need the pull-back operations.

Again, the lack of mathematical rigor is vast.

Response

a) It is confirmed that Eq (2) in the SI section is a rate-dependent equation. Our intent in stating it was to provide a mathematical expression that represents system behaviour as described immediately preceding it (not repeated here for the sake of brevity).

The multiscale modelling approach presented in the manuscript leads to a model where none of the component hypomodels and relation models comprising it contain a rate-dependent constitutive law. These models are validated by experimental evidence (see response to Comment R3.10). Hence, concerns raised regarding integrability and assurance of frame invariance are inconsequential.

b) In Ju [34] p. 2513, the author gives the following rate-dependent equation in running form in the sentence immediately following Eq (32) in that paper:

"S" ˙"=" "A" ^"ep" ":" "E" ˙

There "S" , "A" ^"ep" and "E" are identified with the second PK stress, the elastoplastic damage tangent modulus and the GL strain respectively. These definitions are identical to our definitions of σ, Hepd and E respectively in Eq (2). We have now cited Ju [34] on line 112 of the revised Supporting Information section in support for the expression we used in Eq (2).

c) The abstraction of reality as described in the SI section, and the multiscale model that follows from it, suggest that the determination of the elastoplastic damage modulus tensor Hepd is unnecessary. This was already stated in SI §S2.8 and is quoted again below for reference:

“Note that the final result does not depend on the intermediate solution variables Hepd[1] and Hepd[2]. Hence, one may choose to not solve equations (S1-i) and (S2-i) without affecting the result.”

This should not at all be construed to imply that the present approach is somehow limited in that it will always lead to multiscale models where such sophistication is avoided. Indeed, one may modify the abstraction of reality in at least two different ways such that the elastoplastic damage modulus tensor will need to be determined. One is where the effect of mechanoregulation on determining future femur strength is considered important. Another is where the modeller hypothesizes that including elastoplastic damage process will predict femur strength more accurately than in the present abstraction.

In such cases, the question posed by the reviewer becomes relevant; however, the resolution is straightforward. The modeller will simply specify mathematical forms for the moduli Hepd[1] and Hepd[2] in the component hypomodels at scales S1 and S2 in a way that agrees with experimental observations, and validate the component hypomodels against experiments conducted separately at each scale (see response to Comment R3.10). The form of the relation model R12 between these scales will be validated by comparing predictions against combined experimental data taken at the two scales. As such, ensuring mathematical consistency between these models at different scales is not needed.

The manuscript describes a method to develop a multiscale model for a general multiscale problem. It is not our intention to propose one specific model for a specific problem. This generality of intent is the reason our mathematical expressions are general. We do not preclude the application of mathematical rigour where it is useful.

Comment R3.10

9) The manuscript lacks any concepts of verification and validation in computational science and engineering. Please see Patrick J. Roache, Verification and validation in computational science and engineering, Hermosa, 1998.

Response

We thank the reviewer for this excellent remark. There is a large body dedicated to credibility of computational models, which includes verification, validation, uncertainty quantification (VVUQ) and context of use (CoU). Some recent papers are now referenced the revised manuscript [35-41]. These point to the fact that there remains some debate regarding the definitions of the various aspects of credibility (VVUQ and CoU). Briefly, the definitions depend on whether the model is an instrument used to establish new knowledge or to solve a problem. This separation (albeit not a clear one) maps to a separation between fields of science (physics, biology) on one hand and applied fields such as engineering and medicine on the other. The methods for assuring model credibility, i.e. performing VV&UQ for a given CoU, also differ based on which definitions are used [39-41]. It is not our intent to limit the future application of the multiscale modelling approach presented here to any specific use (seek knowledge vs solve problem) or area (science vs engineering and medicine). Hence, the present approach does not prefer any specific definition for VV&UQ and CoU. It is also our view that no new definition or method for VV&UQ and CoU is necessitated by the present approach.

A process for credibility assessment of multiscale models, as developed using the present approach, is summarised on lines 626–693 and the new Fig 7A of the revised manuscript. Below, we illustrate the application of some elements of this process ¬– specifically, validation and context of use specification – by considering a multiscale model for 10 year femur strength as created using the present approach. As the revised manuscript details, in the present approach component “real” scales are defined based on experiment capabilities. This readily allows for the possibility of validating the corresponding component hypomodels using experimental data. Even so, the modeller must acknowledge the not entirely mitigable limitation of this validation exercise: that the experiment conducted at any component “real” scale is not the same system where the multiscale model will finally be applied. For example, CT based finite element (FE) models of femur strength can only be validated on cadaveric femurs (“validation context”) because femur strength measurement is destructive. To get around this problem when applying CT-based FE models in the in vivo context, some authors have sought further validation by quantifying the accuracy of stratifying fracture status (observable in vivo) in a retrospective cohort based on the model predicted femur strength [13, 14, 16]. However, if these models are to be used to predict femur strength in vivo (scale S1 in main text, “application context”), the differences with respect to the validation context must be acknowledged (fracture status in retrospective cohort vs femur strength in prospective cohort). Such limitations are rarely fully mitigable, and in general reduce the credibility of any model (see also the discussion on CoU below).

The revised manuscript details also the process of validating a relation model. This requires isolating a “smaller” hypothetical scale and physically realising it in experiments. Again, this hypothetical scale is different from the corresponding portion of system where the multiscale model will finally be applied. This is another unmitigable limitation that the modeller needs to acknowledge. For example, consider the multiscale model for predicting 10-year femur strength in a living human described in SI §S2. Consider in addition that the effect of mechanoregulation is non-negligible for the population on whom the final multiscale model will ultimately be applied. This would require additional relation model(s) to predict bone cellular activity signal ("s" _"k,α" ^"h" ) at scale S1 in dependence of external loads on the whole femur (τ[3]) at scale S3. Now, consider that the component hypomodels S1 and S2 have been validated with respect to separate experiments at corresponding scales. The modeller constructs a model sub-system spanning scales S1 and S2, but additionally ensuring that the effect of mechanoregulation is either negligible or controlled for (and preferably known). The relation model R12 is validated by comparing predictions of the multiscale model comprising S1, S2 and R12 against experimental observations from this model system. As our definition of scale implies, these experiments cannot be combined in a manner such that it is possible to measure features of interest whose characteristic sizes fall anywhere within the spatiotemporal span of scales S1 and S2 taken together. However, this should not be considered a limitation in our view, as it is precisely for this reason that the relation model is needed. In other words, model R12 contains empirical knowledge regarding the system. Moreover, in conducting experiments on the model sub-system, one need not measure any variable that appears as the input of a relation model but as the output of a component hypomodel or vice versa. This relaxation in requirement can be leveraged when designing the experiments. Moving forwards, the modeller validates each relation model by constructing, where necessary, model sub-systems encompassing the spatiotemporal span of the linked component scales taken together.

Once all component hypomodels and relation models are validated, the full multiscale model is validated by comparing the model prediction against observation from experiments conducted at each scale in the application CoU. Regarding CoU, we considered above a (rather typical) case where a model’s validation context differs from its application context, thereby adversely affecting its credibility. In the following, an example is presented to demonstrate that ignoring the CoU can mislead one to believe that AM&CM methods are readily applicable in the biomedical context.

Consider the problem of determining the stiffness of bone tissue regions (both cortical and cancellous bone) within an adult human femur. In the first instance, consider an ex vivo context where the extent-to-grain ratio of available measurement techniques is usually much higher than in vivo. In order to push the narrative in favour of AM&CM methods as far as possible, let us assume (rather liberally) that it is possible to characterise in an excised femur material and geometry features ranging continuously from 10–9 m (feature: tropocollagen molecule diameter; method: neutron diffraction) to 10–2 m (feature: specimen size; method: mechanical testing) in size. Let us assume (again, rather liberally) that a mathematical model is known, which when integrated, will accurately predict the mechanics of bone irrespective of size. This is the “fine-scale” model in the sense of Fish [28]. Consider that using state-of-the-art AM&CM based approaches, the fine scale model can be reduced – in a mathematically rigorous and thermodynamically consistent manner – to a coarse-grained model or indeed a series of such models [see for example 3, 10, 17-21, 42]. This set of successively coarse-grained models is an AM&CM type multiscale model. Let the predictions of such a multiscale model closely match observations from cadaver experiments i.e. we consider the multiscale model validated. Let us also ignore all the limitations of such a model (acknowledged in the above references).

Now, let us consider applying the above “validated” model in the in vivo context. As most of the data needed to inform the model cannot be obtained from a living human, some sort of average data from a cadaver population needs to be used instead. This will lead to higher uncertainty in input than in the original ex vivo context. This will lead in turn to higher inaccuracy in the AM&CM based model output. Due to the highly nonlinear nature of uncertainty propagation, there is no assurance that prediction inaccuracy will not increase further as this tissue mechanics model is coupled with a model to predict whole femur fall strength in vivo. In comparison, a simple regression-based phenomenological model such as that of Morgan et al. [7], when coupled with a whole femur fall strength model, accurately stratifies fracture risk in living humans [15]. This high accuracy is due to the fact that the Morgan et al. [7] model requires as input the volumetric bone mineral density measured using clinical qCT. This measurement is site- and subject-specific and has similar uncertainty in both in vivo and ex vivo contexts. As such, the model of Morgan et al. [7] is arguably much more “useful” in the in vivo context than the models constructed using an AM&CM based approach. Nothing suggests that this difference in usefulness will not increase further as the models are coupled with a model simulating loss of bone with time to predict future bone strength. In summary, it is extremely important to identify the appropriate CoU for a multiscale model when discussing its credibility, and ultimately, its usefulness.

Comment R3.11

10) What is the numerical method used? If the finite element method, what type of element is used? How if the model discretized? Was solution verification performed?

Response

We thank the reviewer for the suggestion.

With regard to the exemplar multiscale model presented in §2.3, the numerical method employed in the S1-scale model is the FE method. The use of the FE method in the context of predicting femur strength has been described extensively in past studies [4, 8, 9, 13-16]. A quadratic tetrahedral mesh with an average element size of 3 mm is used, which is sufficient to predict local strains on the femur neck with an accuracy of ~7% compared to strain gauge measurements in experimental studies. Based on a strain-based failure criterion, these strains are used to estimate the minimum force needed to break the femur (femur strength), the accuracy of which has been shown to be ~15%. This level of accuracy is at par with other state-of-the-art qCT-based FE models of femur side-fall strength [15]. As mentioned in §2.3.8, no further numerical approximation is necessary for models R12 and S2. These details are briefly revisited in lines 641–650 of the revised manuscript.

With regard to the modified multiscale model described in §2.4, the numerical methods employed at the S1-scale is the same as above. The S2�-scale model is an algebraic operation (addition) and the three equations comprising R12� are assignment operations. Hence no further numerical approximation is necessary.

With regard to the multiscale models presented in SI §S2 and §S3, the responses to preceding comments explain why any particular mathematical form was not provided to the equations in these models. Since the mathematical form is not specified, the details of the numerical methods, discretization and solution verification, which depend on this specification, are also omitted.

 

References

1. Yang L, Udall WJ, McCloskey EV, Eastell R. Distribution of bone density and cortical thickness in the proximal femur and their association with hip fracture in postmenopausal women: a quantitative computed tomography study. Osteoporos Int. 2014;25(1):251-63. Epub 2013/05/31. doi: 10.1007/s00198-013-2401-y. PubMed PMID: 23719860.

2. Drugan WJ, Willis JR. A micromechanics-based nonlocal constitutive equation and estimates of representative volume element size for elastic composites. Journal of the Mechanics and Physics of Solids. 1996;44(4):497-524. doi: 10.1016/0022-5096(96)00007-5.

3. Hellmich C, Barthélémy J-F, Dormieux L. Mineral–collagen interactions in elasticity of bone ultrastructure – a continuum micromechanics approach. European Journal of Mechanics - A/Solids. 2004;23(5):783-810. doi: 10.1016/j.euromechsol.2004.05.004.

4. Schileo E, Taddei F, Malandrino A, Cristofolini L, Viceconti M. Subject-specific finite element models can accurately predict strain levels in long bones. J Biomech. 2007;40(13):2982-9. Epub 2007/04/17. doi: 10.1016/j.jbiomech.2007.02.010. PubMed PMID: 17434172.

5. Lees S, Ahern JM, Leonard M. Parameters influencing the sonic velocity in compact calcified tissues of various species. J Acoust Soc Am. 1983;74(1):28-33. Epub 1983/07/01. doi: 10.1121/1.389723. PubMed PMID: 6886195.

6. Ashman RB, Cowin SC, Van Buskirk WC, Rice JC. A continuous wave technique for the measurement of the elastic properties of cortical bone. Journal of Biomechanics. 1984;17(5):349-61. doi: 10.1016/0021-9290(84)90029-0.

7. Morgan EF, Bayraktar HH, Keaveny TM. Trabecular bone modulus-density relationships depend on anatomic site. J Biomech. 2003;36(7):897-904. Epub 2003/05/22. doi: 10.1016/s0021-9290(03)00071-x. PubMed PMID: 12757797.

8. Taddei F, Martelli S, Reggiani B, Cristofolini L, Viceconti M. Finite-element modeling of bones from CT data: sensitivity to geometry and material uncertainties. IEEE Trans Biomed Eng. 2006;53(11):2194-200. Epub 2006/11/01. doi: 10.1109/TBME.2006.879473. PubMed PMID: 17073324.

9. Taddei F, Schileo E, Helgason B, Cristofolini L, Viceconti M. The material mapping strategy influences the accuracy of CT-based finite element models of bones: an evaluation against experimental measurements. Med Eng Phys. 2007;29(9):973-9. Epub 2006/12/16. doi: 10.1016/j.medengphy.2006.10.014. PubMed PMID: 17169598.

10. Hellmich C, Ulm FJ, Dormieux L. Can the diverse elastic properties of trabecular and cortical bone be attributed to only a few tissue-independent phase properties and their interactions? Arguments from a multiscale approach. Biomech Model Mechanobiol. 2004;2(4):219-38. Epub 2004/04/01. doi: 10.1007/s10237-004-0040-0. PubMed PMID: 15054639.

11. Turner CH, Cowin SC, Rho JY, Ashman RB, Rice JC. The fabric dependence of the orthotropic elastic constants of cancellous bone. Journal of Biomechanics. 1990;23(6):549-61. doi: 10.1016/0021-9290(90)90048-8.

12. Rho JY, Hobatho MC, Ashman RB. Relations of mechanical properties to density and CT numbers in human bone. Medical Engineering & Physics. 1995;17(5):347-55. doi: 10.1016/1350-4533(95)97314-f.

13. Falcinelli C, Schileo E, Balistreri L, Baruffaldi F, Bordini B, Viceconti M, et al. Multiple loading conditions analysis can improve the association between finite element bone strength estimates and proximal femur fractures: a preliminary study in elderly women. Bone. 2014;67:71-80. Epub 2014/07/12. doi: 10.1016/j.bone.2014.06.038. PubMed PMID: 25014885.

14. Qasim M, Farinella G, Zhang J, Li X, Yang L, Eastell R, et al. Patient-specific finite element estimated femur strength as a predictor of the risk of hip fracture: the effect of methodological determinants. Osteoporosis International. 2016;27(9):2815–22 doi: 10.1007/s00198-016-3597-4.

15. Viceconti M, Qasim M, Bhattacharya P, Li X. Are CT-Based Finite Element Model Predictions of Femoral Bone Strength Clinically Useful? Curr Osteoporos Rep. 2018;16(3):216-23. Epub 2018/04/16. doi: 10.1007/s11914-018-0438-8. PubMed PMID: 29656377; PubMed Central PMCID: PMCPMC5945796.

16. Altai Z, Qasim M, Li X, Viceconti M. The effect of boundary and loading conditions on patient classification using finite element predicted risk of fracture. Clin Biomech (Bristol, Avon). 2019;68:137-43. Epub 2019/06/16. doi: 10.1016/j.clinbiomech.2019.06.004. PubMed PMID: 31202100.

17. Hellmich C, Ulm F-J. Drained and Undrained Poroelastic Properties of Healthy and Pathological Bone: A Poro-Micromechanical Investigation. Transport in Porous Media. 2005;58(3):243-68. doi: 10.1007/s11242-004-6298-y.

18. Fritsch A, Hellmich C, Dormieux L. Ductile sliding between mineral crystals followed by rupture of collagen crosslinks: Experimentally supported micromechanical explanation of bone strength. Journal of Theoretical Biology. 2009;260(2):230-52. doi: 10.1016/j.jtbi.2009.05.021. PubMed PMID: WOS:000274798300006.

19. Eberhardsteiner L, Hellmich C, Scheiner S. Layered water in crystal interfaces as source for bone viscoelasticity: arguments from a multiscale approach. Comput Methods Biomech Biomed Engin. 2014;17(1):48-63. Epub 2012/05/09. doi: 10.1080/10255842.2012.670227. PubMed PMID: 22563708; PubMed Central PMCID: PMCPMC3877913.

20. Morin C, Hellmich C. A multiscale poromicromechanical approach to wave propagation and attenuation in bone. Ultrasonics. 2014;54(5):1251-69. Epub 2014/01/25. doi: 10.1016/j.ultras.2013.12.005. PubMed PMID: 24457030.

21. Morin C, Vass V, Hellmich C. Micromechanics of elastoplastic porous polycrystals: Theory, algorithm, and application to osteonal bone. International Journal of Plasticity. 2017;91:238-67. doi: 10.1016/j.ijplas.2017.01.009.

22. Scheiner S, Pivonka P, Hellmich C. Coupling systems biology with multiscale mechanics, for computer simulations of bone remodeling. Computer Methods in Applied Mechanics and Engineering. 2013;254:181-96. doi: 10.1016/j.cma.2012.10.015.

23. Colloca M, Blanchard R, Hellmich C, Ito K, van Rietbergen B. A multiscale analytical approach for bone remodeling simulations: linking scales from collagen to trabeculae. Bone. 2014;64:303-13. Epub 2014/04/10. doi: 10.1016/j.bone.2014.03.050. PubMed PMID: 24713194.

24. Pastrama MI, Scheiner S, Pivonka P, Hellmich C. A mathematical multiscale model of bone remodeling, accounting for pore space-specific mechanosensation. Bone. 2018;107:208-21. Epub 2017/11/25. doi: 10.1016/j.bone.2017.11.009. PubMed PMID: 29170108.

25. Hellmich C, Kober C, Erdmann B. Micromechanics-based conversion of CT data into anisotropic elasticity tensors, applied to FE simulations of a mandible. Ann Biomed Eng. 2008;36(1):108-22. Epub 2007/10/24. doi: 10.1007/s10439-007-9393-8. PubMed PMID: 17952601.

26. Blanchard R, Morin C, Malandrino A, Vella A, Sant Z, Hellmich C. Patient-specific fracture risk assessment of vertebrae: A multiscale approach coupling X-ray physics and continuum micromechanics. Int J Numer Method Biomed Eng. 2016;32(9). Epub 2015/12/17. doi: 10.1002/cnm.2760. PubMed PMID: 26666734.

27. Bhattacharya P, Manoharan MP, Govindarajan R, Narasimha R. The critical Reynolds number of a laminar incompressible mixing layer from minimal composite theory. Journal of Fluid Mechanics. 2006;565. doi: 10.1017/s0022112006002047.

28. Fish J. Bridging the scales in nano engineering and science. Journal of Nanoparticle Research. 2006;8(5):577-94. doi: 10.1007/s11051-006-9090-9.

29. Geers MGD, Kouznetsova VG, Brekelmans WAM. Multi-scale computational homogenization: Trends and challenges. Journal of Computational and Applied Mathematics. 2010;234(7):2175-82. doi: 10.1016/j.cam.2009.08.077.

30. Hansen U, Zioupos P, Simpson R, Currey JD, Hynd D. The effect of strain rate on the mechanical properties of human cortical bone. J Biomech Eng. 2008;130(1):011011. Epub 2008/02/27. doi: 10.1115/1.2838032. PubMed PMID: 18298187.

31. Courtney AC, Wachtel EF, Myers ER, Hayes WC. Effects of loading rate on strength of the proximal femur. Calcif Tissue Int. 1994;55(1):53-8. Epub 1994/07/01. doi: 10.1007/BF00310169. PubMed PMID: 7922790.

32. Jazinizadeh F, Mohammadi H, Quenneville CE. Comparing the fracture limits of the proximal femur under impact and quasi-static conditions in simulation of a sideways fall. J Mech Behav Biomed Mater. 2020;103:103593. Epub 2020/02/25. doi: 10.1016/j.jmbbm.2019.103593. PubMed PMID: 32090922.

33. Courtney AC, Wachtel EF, Myers ER, Hayes WC. Age-related reductions in the strength of the femur tested in a fall-loading configuration. J Bone Joint Surg Am. 1995;77(3):387-95. Epub 1995/03/01. doi: 10.2106/00004623-199503000-00008. PubMed PMID: 7890787.

34. Ju JW. Energy‐Based Coupled Elastoplastic Damage Models at Finite Strains. Journal of Engineering Mechanics. 1989;115(11):2507-25. doi: 10.1061/(asce)0733-9399(1989)115:11(2507).

35. Roache PJ. Perspective: Validation—What Does It Mean? Journal of Fluids Engineering. 2009;131(3). doi: 10.1115/1.3077134.

36. Groen D, Richardson RA, Wright DW, Jancauskas V, Sinclair R, Karlshoefer P, et al. Introducing VECMAtk - Verification, Validation and Uncertainty Quantification for Multiscale and HPC Simulations. Computational Science – ICCS 2019. Lecture Notes in Computer Science2019. p. 479-92.

37. National Research Council. Assessing the Reliability of Complex Models. Washington, DC: The National Academies Press; 2012.

38. Hoekstra A, Chopard B, Coveney P. Multiscale modelling and simulation: a position paper. Philos Trans A Math Phys Eng Sci. 2014;372(2021). Epub 2014/07/02. doi: 10.1098/rsta.2013.0377. PubMed PMID: 24982256.

39. Bhattacharya P, Viceconti M. Multiscale modeling methods in biomechanics. Wiley Interdiscip Rev Syst Biol Med. 2017;9(3). doi: 10.1002/wsbm.1375. PubMed PMID: 28102563; PubMed Central PMCID: PMCPMC5412936.

40. Patterson EA, Whelan MP. A framework to establish credibility of computational models in biology. Prog Biophys Mol Biol. 2017;129:13-9. Epub 2016/10/06. doi: 10.1016/j.pbiomolbio.2016.08.007. PubMed PMID: 27702656.

41. Pathmanathan P, Gray RA. Validation and Trustworthiness of Multiscale Models of Cardiac Electrophysiology. Front Physiol. 2018;9:106. Epub 2018/03/03. doi: 10.3389/fphys.2018.00106. PubMed PMID: 29497385; PubMed Central PMCID: PMCPMC5818422.

42. Hamed E, Lee Y, Jasiuk I. Multiscale modeling of elastic properties of cortical bone. Acta Mechanica. 2010;213(1-2):131-54. doi: 10.1007/s00707-010-0326-5.

---

## [Decision Letter · Decision Letter 1]

2 Mar 2021

PONE-D-20-14710R1

A systematic approach to the scale separation problem in the development of multiscale models

PLOS ONE

Dear Dr. Bhattacharya,

Thank you for submitting your manuscript to PLOS ONE. After careful consideration, we feel that it has merit but does not fully meet PLOS ONE’s publication criteria as it currently stands. Therefore, we invite you to submit a revised version of the manuscript that addresses the points raised during the review process.

My own assessment is that while there are weaknesses in the model, as detailed by Reviewer 3, there are also new contributions, noted by Reviewers 1 and 4, that should be put forth for consideration in the open scientific literature, consistent with PLOS ONE's publication policy. Please note that Reviewer 4 was solicited after Reviewer 2 declined to review the revised submission. For this revision, I ask that the authors fully address the new comments raised by Reviewer 4 and also make effort to more carefully address the concerns reiterated by Reviewer 3.

We look forward to receiving your revised manuscript.

Kind regards,

Ryan K. Roeder, PhD

Academic Editor

PLOS ONE

Journal Requirements:

Reviewers' comments:

Reviewer's Responses to Questions

**Comments to the Author**

1. If the authors have adequately addressed your comments raised in a previous round of review and you feel that this manuscript is now acceptable for publication, you may indicate that here to bypass the “Comments to the Author” section, enter your conflict of interest statement in the “Confidential to Editor” section, and submit your "Accept" recommendation.

Reviewer #2: (No Response)

Reviewer #4: (No Response)

2. Is the manuscript technically sound, and do the data support the conclusions?

Reviewer #2: Partly

Reviewer #4: Yes

3. Has the statistical analysis been performed appropriately and rigorously? 

Reviewer #2: N/A

Reviewer #4: N/A

4. Have the authors made all data underlying the findings in their manuscript fully available?

Reviewer #2: No

Reviewer #4: Yes

5. Is the manuscript presented in an intelligible fashion and written in standard English?

Reviewer #2: Yes

Reviewer #4: Yes

6. Review Comments to the Author

Reviewer #2: as the authors' proposition for the scale separation problem is in conflict with major premises of continuum mechanics, a more detailed review was necessary, and is attached as a separate file.

Reviewer #4: This paper addresses an important topic of scale separation in multiscale modeling of materials. However, the reviewer has general comments on how to improve this paper. The literature review is rather limited and overlooking some important topics and works.

1. The scale separation in modeling is certainly an important condition, and it is tacitly assumed in most studies. The idea of “grain” and “extent” is novel, and linking scales to imaging equipment scale is practical and interesting. However, the other important concept closely related but untouched in the paper is the concept of a representative volume element. The term is not mentioned in the paper except for a reference by Drugan and Willis, which is not representative (their results only hold for periodic microstructures). Another important aspect is material randomness, which again is fully overlooked. A comprehensive reference on these three topics (scale separation, representative volume element, and randomness) is, for example, a recent book (A. Malyarenko and M. Ostoja-Starzewski, Tensor-Valued Random Fields for Continuum Physics, Cambridge University Press, 2019), which summarizes past works in this area. The reviewer does not recommend incorporating these concepts in the model, but these topics merit at least mentions.

2. Secondly, the authors are citing in their revised paper about ten papers by the Hellmich group, which were not present in the original version of the article. They do not mention other similar works employing micromechanics approaches. The reviewer recommends that the references on analytical micromechanics formulations are more balanced, not favoring a specific research group.

The model is relatively crude, but the incorporation of both spatial and temporal scales in this simple model is a substantial contribution. Also, the example of a femur modeled over ten years is of high clinical interest. The ideas presented in this paper should stimulate more realistic models of bone and may stimulate research in other areas.

In summary, the reviewer recommends this paper for publication after the points raised above are addressed.

7. PLOS authors have the option to publish the peer review history of their article (what does this mean?). If published, this will include your full peer review and any attached files.

Reviewer #2: No

Reviewer #4: No

---

## [Author Response · Author response to Decision Letter 1]

23 Apr 2021

1. Responses to Reviewer #2 comments

Comment R2.1

As the authors' proposition for the scale separation problem is in conflict with major premises of continuum mechanics, a more detailed review was necessary, and is attached as a separate file [Author’s Note: This is appended below].

The reviewer appreciates that the authors have provided a quite comprehensive response to the questions raised in the first round of reviews, complemented by some additional remarks in the revised manuscript. While still not described in minute detail, the authors seem to roughly perform the following computing steps for the human femur analysis, by combining fitting functions with linear elastic FE analysis:

(i) Calibration of CT-Hounsfield values to volumetric bone mineral density (depicted in Figures 1 and 4);

(ii) probably relating the latter to apparent mass density, in order to then;

[Reviewer #2’s Note A: The authors write that REF 32 Altai et al would give information on CT data processing, however, the reviewer could not even find the abbreviation “CT” in this paper]

(iii) use REF 41-Morgan et al’s fitting relationship between apparent mass density and loading modulus;

(iv) use of corresponding modulus values in linear elastic Finite Element analysis, in order to compute principal strains;

(v) averaging the strain values over pre-set regions of interest, and

(vi) inserting the latter into a strain-based failure criterion.

[Reviewer #2’s Note B: It would really helpful to state this more explicitly in the manuscript, in the context of Eq.(5).]

In their response, the authors stress that this approach was validated in Taddei et al, 2007, Med Eng Phys 29, 973-979, 2007; however, the latter reference uses Keller’s 1994 [Reviewer #2’s Note C: J Biomech 1994; 27(9):1159-68] density-to-modulus relationship (rather than the relationship of Morgan et al) and obviously ex vivo experiments. In this context, the reviewer, being asked for a rigorous review, re-iterates on several, so far hardly addressed critical scientific issues, which – in one way or another – were already referred to in the first round of reviews.

Response

The reviewer has correctly outlined the FE analysis approach to predict bone strength. 

However, we would like to point out that in §1 Introduction of Altai et al (2019), the last sentence of the first paragraph reads: “The most investigated alternative method is the quantitative-computed-tomography (QCT)-based subject-specific finite element modelling (FE) (Viceconti et al., 2018)”. There are three other mentions of the abbreviation CT in the main text of the article (excluding references).

More importantly, the first paragraph of §2.2 Finite element modelling in Altai et al. (2019) explains how the FE model is obtained from CT scans. As this is a well-established methodology, hence Altai et al (2019) refer to past papers when explaining the method. This FE modelling method is not the novelty of the present manuscript. Hence, we refer the reader to Altai et al (2019) (Ref 31 in the present manuscript) for the details. We believe that repeating these details could potentially distract a reader from the main message of our multiscale modelling approach. Following the reviewer’s suggestion, on lines 246–247 we have clarified the use of the strain-based failure criterion in Eq. (1) (we believe “Eq. (5)” in the comment is a misprint as that equation describes the change in bone strength over the 10-year period).

In response to Comment R2.2 in the last revision, it was stated: “In the in vivo setting, multiple studies [13-16] have shown that the regression model of Morgan et al. [7] coupled with an FE modelling approach based on Taddei et al. [9] predicts bone strength values that can classify fracture status very accurately.” To imply from this statement that Taddei et al used the regression result of Morgan et al is a misunderstanding.

Comment R2.2

1. Mass-density-to-modulus fitting functions reflecting corresponding experimental data on human femoral bone show very large variations, and differ from each other by several hundred percent; as reviewed in Clin Biomech 23, 135–146, 2008, see in particular Figure 4(a).

Response

Helgason et al (Clin Biomech 23, 135–146, 2008) point out that for a given anatomical site, the measurement of mechanical response is affected by experimental conditions e.g. specimen size and shape and end support. As the reviewer points out, this is the uncertainty in the measurement data, and should not be confused with prediction accuracy of a given regression model.

Helgason et al note that a cylindrical sample with 2:1 aspect ratio analysed using the end-cap technique leads to the highest accuracy and precision. Once all these experimental conditions are met – as in the studies of Morgan et al (2003) – one can assess the error in using the regression model to predict the mechanical response. As mentioned in response to Comment R2.2 (previous revision), this error is ~ 443 MPa for Morgan et al (2003) fitting function, and smaller than that of the Hellmich et al (2003) model. The bone density to elasticity relationship used in the present study is that of Morgan et al (2003); this is already specified on line 242 of the manuscript and papers cited therein (Refs 40 and 41 in particular).

As Comment R2.2 concerns only our response in the last revision, and not any statement in the manuscript itself, no corresponding changes to the manuscript have been made.

Comment R2.3

2. The aforementioned moduli were determined in loading mode. This entails a critical issue concerning elasticity in the sense of continuum mechanics and thermodynamics, see e.g. well-known landmark contributions such as Coleman and Noll, The Foundations of Mechanics and Thermodynamics, pp. 145-156, 1974; Coussy, Poromechanics, Wiley, 2004; Rajagopal, Z Angew Math Phys 58, 309–317, 2007, and references therein: elasticity is associated with the conversion of internal energy into efficient mechanical work. Mechanical work is only gained during the unloading phase, which is indeed characterized by higher (truely elastic) moduli, see e.g. J Mech Beh Biomed Mat 52, 51-62, 2015. Hence, the authors’ comparison, in response to Comment R2.2, of fitted loading moduli with micromechanics-predicted elasticity tensor components (validated by the fastest ultrasonics waves, which are always associated with elasticity, see e.g. Carcione, Wave Fields in Real Media, Oxford, 2001) lacks scientific rigour.

Response

Our response to Comment R2.2 (previous revision) responded partly to the suggestion in Comment R2.1 (previous revision) that multiscale problems in bone biomechanics have already been “largely solved”. Thus, we identified the problem of predicting bone strength over a 10-year period. The articles suggested by the reviewer in Comment R2.2 (previous revision) contained several models for bone tissue mechanics. At the point of writing the response, we were unaware of a FE modelling pipeline that employs micromechanics-based elasticity tensors (please see Comment R2.8 below and our response to the same). Hence, to further the reviewer’s argument, we inserted the micromechanics-based elasticity tensor components into our finite element modelling pipeline which employs fitted moduli measured in loading mode. Our response did not seek to compare fitted loading moduli with micromechanics-predicted elasticity tensor components, but comparison of errors in FE predictions based on these. Our analysis suggested that “it is doubtful that if the model of Hellmich et al. [3] was used to predict femur strength, the resulting accuracy would be higher than that of the S1 scale model of §2.3.” On this basis, we differed from the reviewer’s opinion that there is no need for a new multiscale modelling approach as multiscale problems in bone biomechanics have been “largely solved”.

The above conclusion remains unchanged upon reanalysing the situation in view of the papers kindly pointed out by the reviewer. Details are given in response to Comment R2.8.

As Comment R2.3 concerns only our response in the last revision, and not any statement in the manuscript itself, no corresponding changes to the manuscript have been made.

Comment R2.4

3. The authors unarguably use the scientific language and mathematical formalisms of continuum mechanics (e.g. Cauchy stress tensor and elastic stiffness tensor on page 10), and at the same time, they write, in response to Comment R2.4, “the approach presented in this manuscript should in no way be confused with the AM&CM” (standing for applied mathematics and continuum mechanics). Already from a purely logical viewpoint, this does not make sense.

Response

Applied Mathematics and Continuum Mechanics (AM&CM) is a label. No label can fully explain what the scientific precepts of a method are (or are not). Formal definitions of the collection of multiscale modelling approaches to which AM&CM refers can be found in papers such as Fish (2006) and Geers et al (2010) (Refs 9 and 10 in the manuscript). In addition, throughout the manuscript we define and describe our multiscale modelling approach and discuss in detail how it differs from the AM&CM approach. As such, we clearly state in line 110–112 that we expect the present multiscale modelling approach to be applied largely in situations where continuum mechanics is the only appropriate physical idealisation. It is therefore our opinion that in to understand what the scientific precepts of the current approach are, a reader will not need to draw any inferences from the “AM&CM” label itself (or even any parts or permutations of it).

As Comment R2.4 concerns only our response in the last revision, and not any statement in the manuscript itself, no corresponding changes to the manuscript have been made.

Comment R2.5

4. More importantly, the stress tensor, as originally introduced by Cauchy in the 1820s, is connected to forces acting centrally on facets of a small solid element (as can be found in virtually any textbook on continuum mechanics, see e.g. Salençon, Handbook of Continuum Mechanics, Springer, 2001; Fung et al, Classical and Computational Mechanics, 2017), and it was not before 1941, that a satisfying extension of the stress tensor concept towards porous materials was proposed, namely by M.A.Biot, first in the context of soils (quote from Journal of Applied Physics 12, 155ff. 1941): “Consider a small cubic element of the consolidating soil, its sides being parallel with the coordinate axes. This element is taken to be large enough compared to the size of the pores so that it may be treated as homogeneous, and at the same time small enough, compared to the scale of the macroscopic phenomena in which we are interested, so that it may be considered as infinitesimal in the mathematical treatment.” Similar definitions were given in the framework of continuum micromechanics in the 1960s. Hence, the reviewer is not aware of any well-accepted historical or conceptual grounds based on which the stress tensor may be applied to porous materials (such as bone), other than elements similar to the aforementioned “small cubic element”, i.e. RVEs in the currently dominating terminology. Thanks to the seminal work of Cowin and colleagues, see e.g. J Biomech 32, 217-238, 1999, this reasoning has become a major theoretical fundament in bone biomechanics as well.

Response

We agree fully with this comment. Indeed, we are operating under the same mathematical and mechanical precepts. The divergence is in the application of these concepts, as detailed in the response to the following comment. The manuscript remains unchanged in response to this particular comment.

Comment R2.6

5. Beyond that, and practically speaking, the use of Cauchy’s stress tensor has major consequences for the scale of instrumentation l* (a central concept put forward by the authors), both in ultrasonic tests and mechanical tests, as discussed next.

6. According to page 6 of the manuscript, “one can ignore variations separated by distances l* or smaller.” Similarly, on page 3 of the supplementary material, one can read “for any two points that are separated by a distance smaller than l*, variables of interest taken at these points (e.g. location, stress, traction, displacement and strain) cannot be distinguished from each other.” In other words, at the scale l*, the quantities do not vary with space - they are homogeneous. Accordingly, scale l* in ultrasonic tests is not the sample size (as assumed by the authors when deploring “the discrepancy of scales compared by Hellmich et al [3]” in their response to Comment R2.2), but is governed by the fluctuation length / wavelength, see (again) e.g. J Eng Mech 128, 808-816, 2002 or J Theor Biol 244, 597–620, 2007 for the requirement l << L, which needs to hold for any continuum formulation (including that of the authors). Only for lengths sufficiently smaller than the wavelength, the scale l* is reached. Accordingly, MHz-frequency tests typically relate to material scales l* (RVE-sizes) much smaller than the ultrasonic sample size. Conclusively, one sees that the instrumentation scale l* should definitely not be arbitrarily chosen, but needs to comply with the employed physical/mechanical premises (e.g. equations of motions, plane waves, Cauchy stress etc.).

7. The situation with mechanical test is similar. If homogeneous traction forces are prescribed on the boundaries of a sample (as in carefully designed uniaxial testing), the sample size is indeed l*. In case of bending tests with stress gradients, as done by Morgan et al, l* needs to be sufficiently small with respect to structural loading scale (sigma / grad sigma), so that the l* (and correspondingly tested RVE) is again smaller than the sample size.

Response

Point #5 can be alternatively restated as: the use of Cauchy’s stress tensor has major consequences on the accuracy of the model when the scale is fixed by the choice of instrumentation. This is our preference, as it allows to choose the experiment arbitrarily, to quantify the error in prediction (which are bound to result from neglecting the requirements mentioned in #6 and #7), and to propagate this error through the model – all methodologically rigorous and scientifically valid steps. The error is quantified by comparing model prediction with observation at each component scale. Verification, validation and uncertainty quantification of the present approach are fully detailed in lines 635–712 of the manuscript.

Our preference as mentioned above is motivated as follows. Even for the most carefully constructed experiment, conditions of “homogeneity” are attained in the limit. Deviations from this condition always exist, even though these are finitely small. We argue that constraining the instrumentation (while carefully constructing the experiment) can limit the scenarios in which the resulting multiscale model can be applied.

As Comment R2.6 concerns only our response in the last revision, and not any statement in the manuscript itself, no corresponding changes to the manuscript have been made.

Comment R2.7

8. In contrast to the authors statements on line 80 of the revised manuscript, all continuum micromechanics models discussed by the authors do not imply time and space periodicity. The authors need to distinguish between far-field based homogenization approaches (without any type of periodicity required) and asymptotic homogenization approaches.

Response

We have amended the statements on lines 72, 79, 85, 113 and 116 in the revision to include macrohomogeneity as the basis for mean field homogenization approaches.

Comment R2.8

9. Somehow at odds with the authors’ statements on the open literature, CT-informed coupled X-ray-physics-micromechanics models were applied to the human femur problem and validated against experiments on whole femurs, see e.g. Int J Multiscale Comp Eng 6, 483-498, 2008; J Biomech Eng. 133, 061001, 2011; and corresponding elastoplastic approaches for vertebra bones have been shown to agree with in vivo fracture cases observed in athletes, see Int J Numer Meth Biomed Engng 32, e02760, 2016.

The reviewer has given these statements for the sake of a transparent scientific discussion. The authors may find these remarks useful when addressing a larger scientific community, going beyond the confines of traditional bone biomechanics. Needless to say, any decisions on the suitability of the manuscript for PLoS ONE are entirely in the hands of the handling editor.

Response

In the previous revision the reviewer kindly referred us to some papers by Hellmich et al. as examples of micromechanics (MM)-based multiscale modelling approaches. The main argument was that these works have to a great extent solved the multiscale challenge in bone mechanics. In our response to that Comment R2.2 (previous version) we raised a doubt against this argument – and detailed whether an MM-model (such as detailed in these papers) would lead to a more accurate prediction of femur strength than the empirical model used to derive constitutive properties in the present approach. The doubt was based on the observation that errors in predicting true MM-derived elastic moduli were in excess of errors in predicting the stiffness measured in loading mode using empirical relationships (such as those of Morgan et al). We postulated that this larger uncertainty in quantifying elasticity could potentially balance out improvements due to the consideration of orthotropy in material response as informed by a micromechanics approach.

At that time of writing the previous response, we were indeed unaware of the abovementioned FE analyses of human femur based on MM-derived moduli. We thank the reviewer for pointing out these additional papers by Trabelsi and co-workers. Yet, these papers only confirm our previous conclusion. Although based on a limited study (in all, 5 cadaveric femurs), the more recent article, Trabelsi et al (J Biomech Eng. 133, 061001, 2011) conclude that “The differences between the isotropic model [i.e. empirical model] and the orthotropic MM model [with regard to displacements and strains] are in the range of the experimental errors so unequivocal conclusions cannot be drawn regarding which model is better”. The explanatory remarks in square brackets […] are ours. Note that in the clinical problem considered in the illustrations in the present study, the final objective is to predict femur strength, and the above cited studies conducted by Trabelsi and co-workers do not predict femur strength. As such, MM-based models cannot be taken to be the “final word” on multiscale modelling of femur strength, and there remains an unmet need to develop new multiscale modelling methods.

We hope that a response to the remark regarding models of vertebra is not needed as their performance has little bearing on the femur strength problem.

As Comment R2.8 concerns only our response in the last revision, and not any statement in the manuscript itself, no corresponding changes to the manuscript have been made.

 

2. Responses to Reviewer #4 comments

Comment R4.1

This paper addresses an important topic of scale separation in multiscale modeling of materials. However, the reviewer has general comments on how to improve this paper. The literature review is rather limited and overlooking some important topics and works.

1. The scale separation in modeling is certainly an important condition, and it is tacitly assumed in most studies. The idea of “grain” and “extent” is novel and linking scales to imaging equipment scale is practical and interesting. However, the other important concept closely related but untouched in the paper is the concept of a representative volume element. The term is not mentioned in the paper except for a reference by Drugan and Willis, which is not representative (their results only hold for periodic microstructures). Another important aspect is material randomness, which again is fully overlooked. A comprehensive reference on these three topics (scale separation, representative volume element, and randomness) is, for example, a recent book (A. Malyarenko and M. Ostoja-Starzewski, Tensor-Valued Random Fields for Continuum Physics, Cambridge University Press, 2019), which summarizes past works in this area. The reviewer does not recommend incorporating these concepts in the model, but these topics merit at least mentions.

Response

We thank the reviewer for their comment. We already discussed the term scale separation at length in the manuscript – indeed it is in the title of the manuscript and the focus of the introduction section. In the revised version, we have included a brief discussion of the concepts of representative volume element and material randomness on lines 78–81.

Comment R4.2

2. Secondly, the authors are citing in their revised paper about ten papers by the Hellmich group, which were not present in the original version of the article. They do not mention other similar works employing micromechanics approaches. The reviewer recommends that the references on analytical micromechanics formulations are more balanced, not favoring a specific research group.

The model is relatively crude, but the incorporation of both spatial and temporal scales in this simple model is a substantial contribution. Also, the example of a femur modeled over ten years is of high clinical interest. The ideas presented in this paper should stimulate more realistic models of bone and may stimulate research in other areas.

In summary, the reviewer recommends this paper for publication after the points raised above are addressed.

Response

In the revised version, we cite only one paper by the Hellmich group (Ref 15) and have included another paper from a different research group (Ref 16) on line 85. Although there are several research groups working on analytical micromechanics formulations, we chose these works as they also discuss bone mechanics, in order to set the context for the illustrative examples.

---

## [Editor Report · Decision Letter 2]

26 Apr 2021

A systematic approach to the scale separation problem in the development of multiscale models

PONE-D-20-14710R2

Dear Dr. Bhattacharya,

We’re pleased to inform you that your manuscript has been judged scientifically suitable for publication and will be formally accepted for publication once it meets all outstanding technical requirements.

Thank you for your patience and integrity in what was an unusually long review process. 

Kind regards,

Ryan K. Roeder, PhD

Academic Editor

PLOS ONE
---

## [Editor Report · Acceptance letter]

3 May 2021

PONE-D-20-14710R2 

A systematic approach to the scale separation problem in the development of multiscale models 

Dear Dr. Bhattacharya:

I'm pleased to inform you that your manuscript has been deemed suitable for publication in PLOS ONE. Congratulations! Your manuscript is now with our production department. 

Kind regards, 

on behalf of

Dr. Ryan K. Roeder 

Academic Editor

PLOS ONE